# Impact of COVID-19-related disruptions to measles, meningococcal A, and yellow fever vaccination in 10 countries

Katy AM Gaythorpe[1‡], Kaja Abbas[2‡], John Huber[3‡], Andromachi Karachaliou[4‡], Niket Thakkar[5‡], Kim Woodruff[1], Xiang Li[1], Susy Echeverria-Londono[1], VIMC Working Group on COVID-19 Impact on Vaccine Preventable Disease, Matthew Ferrari[6†], Michael L Jackson[7†], Kevin McCarthy[5†], T Alex Perkins[3†], Caroline Trotter[4†], Mark Jit[2,8†*]

[1]MRC Centre for Global Infectious Disease Analysis, Abdul Latif Jameel Institute for Disease and Emergency Analytics (J-IDEA), School of Public Health, Imperial College London, London, United Kingdom; [2]Centre for Mathematical Modelling of Infectious Diseases, London School of Hygiene & Tropical Medicine, London, United Kingdom; [3]Department of Biological Sciences, University of Notre Dame, South Bend, United States; [4]Department of Veterinary Medicine, University of Cambridge, Cambridge, United Kingdom; [5]Institute for Disease Modeling, Bill & Melinda Gates Foundation, Seattle, United States; [6]Pennsylvania State University, University Park, United States; [7]Kaiser Permanante Washington, Seattle, United States; [8]School of Public Health, University of Hong Kong, Hong Kong SAR, China

**\*For correspondence:**
Mark.Jit@lshtm.ac.uk

†These authors contributed equally to this work

‡These authors also contributed equally to this work

**Group author details:**
VIMC Working Group on COVID-19 Impact on Vaccine Preventable Disease See page 11

## Abstract

**Background:** Childhood immunisation services have been disrupted by the COVID-19 pandemic. WHO recommends considering outbreak risk using epidemiological criteria when deciding whether to conduct preventive vaccination campaigns during the pandemic.

**Methods:** We used two to three models per infection to estimate the health impact of 50% reduced routine vaccination coverage in 2020 and delay of campaign vaccination from 2020 to 2021 for measles vaccination in Bangladesh, Chad, Ethiopia, Kenya, Nigeria, and South Sudan, for meningococcal A vaccination in Burkina Faso, Chad, Niger, and Nigeria, and for yellow fever vaccination in the Democratic Republic of Congo, Ghana, and Nigeria. Our counterfactual comparative scenario was sustaining immunisation services at coverage projections made prior to COVID-19 (i.e. without any disruption).

**Results:** Reduced routine vaccination coverage in 2020 without catch-up vaccination may lead to an increase in measles and yellow fever disease burden in the modelled countries. Delaying planned campaigns in Ethiopia and Nigeria by a year may significantly increase the risk of measles outbreaks (both countries did complete their supplementary immunisation activities (SIAs) planned for 2020). For yellow fever vaccination, delay in campaigns leads to a potential disease burden rise of >1 death per 100,000 people per year until the campaigns are implemented. For meningococcal A vaccination, short-term disruptions in 2020 are unlikely to have a significant impact due to the persistence of direct and indirect benefits from past introductory campaigns of the 1- to 29-year-old population, bolstered by inclusion of the vaccine into the routine immunisation schedule accompanied by further catch-up campaigns.

**Conclusions:** The impact of COVID-19-related disruption to vaccination programs varies between infections and countries. Planning and implementation of campaigns should consider country and infection-specific epidemiological factors and local immunity gaps worsened by the COVID-19 pandemic when prioritising vaccines and strategies for catch-up vaccination.

**Funding:** Bill and Melinda Gates Foundation and Gavi, the Vaccine Alliance.

## Introduction

Childhood immunisation services have been disrupted by the COVID-19 pandemic in at least 68 countries during 2020 with around 80 million under 1-year-old children being affected (*Nelson, 2020*; *Science (AAAS), 2020*; *UNICEF, 2020*; *WHO, 2020a*). This has occurred for several reasons – the diversion of health care staff, facilities, and finances to deal with COVID-19 treatment and response; reluctance of individuals to bring children to be vaccinated due to fear of infection; barriers to travel due to local physical distancing measures; disruptions in vaccine supply chains; lack of personal protective equipment; and decisions to stop or postpone vaccination campaigns to reduce the risk of transmission during such campaigns.

The World Health Organization (WHO) issued guidance in March 2020 on immunisation activities during the COVID-19 pandemic (*WHO, 2020b*). The guidance recommended a temporary suspension of mass vaccination campaigns, but continuation of routine immunisation services by the health systems while maintaining physical distancing and infection prevention and control measures for COVID-19. Routine immunisation was one of the most disrupted services relative to other essential health services based on a WHO pulse survey in May and June 2020 that was focused on continuity of essential health services during the COVID-19 pandemic (*WHO, 2020c*). WHO, UNICEF, Gavi, the Vaccine Alliance, and their partners also conducted two pulse polls in April and June 2020 to understand COVID-19-related disruptions to immunisation services (*WHO, 2020d*). Based on respondents from 82 countries, pulse polls indicated that there was widespread disruption to routine immunisation services in addition to the suspension of mass vaccination campaigns. The main reasons reported for this disruption were low availability of personal protective equipment for healthcare workers, low availability of health workers, and travel restrictions.

Disruptions to routine health care due to the COVID-19 pandemic are projected to increase child and maternal deaths in low-income and middle-income countries (*Roberton et al., 2020*). No country has made a policy decision to stop routine immunisation during a COVID-19 epidemic. Risk-benefit analysis of countries in Africa shows routine immunisation to have far greater benefits than risks even in the context of the COVID-19 pandemic (*LSHTM CMMID COVID-19 Working Group et al., 2020*). Nevertheless, routine immunisation coverage has dropped in most countries (*WHO, 2020d*).

Evidence on the health impact of suspending vaccination campaigns during the COVID-19 pandemic is limited. Modelling indicates that both fixed post and door-to-door campaigns targeting under 5-year-old children may cause temporary minor increases in total SARS-CoV-2 infections (*Hagedorn et al., 2020*). However, avoiding campaigns during the local peak of SARS-CoV-2 transmission is key to reducing the effect size, and SARS-CoV-2 transmission during campaigns can be minimised with good personal protective equipment and limiting movement of vaccinators (*Hagedorn et al., 2020*). The WHO recommends that countries consider the risk of outbreaks using epidemiological criteria when deciding whether to conduct preventive vaccination campaigns during the COVID-19 pandemic, but the guidance was not based on any quantitative assessment of transmission risk for either COVID-19 or existing vaccine-preventable diseases (*WHO, 2020e*).

Hence, countries need to assess the health impact of postponing vaccination campaigns, which can inform the epidemiological risk assessment for outbreaks due to campaign delays and prioritise which vaccines to use in campaigns (*WHO, 2020f*). The need for such assessments is greatest in low- and lower middle-income countries which generally have greater risks of vaccine-preventable disease outbreaks and limited health care resources to deal with COVID-19 epidemics. It is difficult to quantify the impact of different scenarios using only observational data, which does not give the counterfactual to what actually happened in 2020. To address this, we used transmission dynamic models to project alternative scenarios about postponing vaccination campaigns alongside disruption of routine immunisation, for three pathogens with high outbreak potential and for which mass vaccination campaigns are a key delivery mode alongside routine immunisation – measles, meningococcal A, and yellow fever.

## Materials and methods

Deaths and disability-adjusted life years (DALYs) due to measles, meningococcal A, and yellow fever under different routine and campaign vaccination scenarios were projected in a subset of 10 low- and lower middle-income countries over the years 2020–2030. Projections were made using previously validated transmission dynamic models; we used three models for measles, two models for meningococcal A, and two models for yellow fever (summary model details are available in *Table 1a-c* with full model details in Appendix Section 3; a description of the key drivers of similarities and differences between models is given in Appendix Section 4). Guidance used by the different models for DALY calculations are publicly accessible (*Vaccine Impact Modelling Consortium, 2019*) and a glossary of terms can be found in *Appendix 1—table 15*.

The chosen countries were low- and lower-middle-income countries that had planned vaccination campaigns in 2020 and were selected following consultations with partners in WHO, UNICEF, CDC and other organisations. Thereby, the selected countries differ between infections – Bangladesh, Chad, Ethiopia, Kenya, Nigeria, and South Sudan for measles; Burkina Faso, Chad, Niger, and Nigeria for meningococcal A; Democratic Republic of the Congo, Ghana, and Nigeria for yellow fever.

Models used routine and campaign vaccination coverage from WUENIC (WHO and UNICEF Estimates of National Immunization Coverage) and post campaign surveys for 2000–2019 (*Li et al., 2021*), and future projections of routine coverage based on assumptions agreed with disease and immunisation programme experts at the global, regional, and national levels (see *Appendix 1—table 16*). Assumptions for our counterfactual 'business as usual' scenario were determined through consultation with disease and immunisation programme experts across partners at the global, regional, and national levels. All assumptions varied by pathogen. For routine immunisation, assumptions about future coverage levels were based on historical coverage from WUENIC for 2015–19. For vaccination campaigns or supplementary immunisation activities (SIA), assumptions about future campaigns were based either on patterns of past campaigns or campaigns recommended by WHO. We explored four scenarios that assumed different levels of disruption in the year 2020 to routine immunisation and postponement of campaigns projected in the scenarios, due to COVID-19 (see *Table 2*). The disruption scenarios are based on 50% reduction in routine immunisation and/or suspension of campaign vaccination in 2020 and postponement to 2021. These disruption scenarios aimed to approximate plausible drops in routine coverage levels and plausible delays to campaigns due to the COVID-19 pandemic.

We estimated the health impact of these disruption scenarios in comparison to the counterfactual scenario of no disruption (BAU – business-as-usual scenario) for measles, meningococcal A, and yellow fever during 2020–2030. We estimated the health impact of routine and campaign immunisation disruption through projections of total deaths (and DALYs) per 100,000 population, excess deaths (and DALYs) per 100,000 population, and excess deaths (and DALYs) during 2020–2030 which were scaled relative to the maximum number of excess deaths (or DALYs) across all scenarios. We did not assume any changes to case-fatality risks as a result of the COVID-19 pandemic.

The models generally produce a range of stochastic realisations based on distributions of input parameters and/or posterior distributions of fitted parameters. In the results, we present output from an average scenario, which is defined differently across models based on their characteristics: model projection from mean (measles/DynaMICE) or median (YF/Imperial) of input parameters, median projection from posterior of fitted force of infection (YF/Notre Dame), mean of stochastic output projections (measles/IDM, measles/PSU, MenA/Cambridge, MenA/KP).

## Results

The health impact varies across the disruption scenarios for the three infections in the different countries. *Figure 1* shows the model-predicted total deaths per 100,000 population per year during 2020–2030 (see *Appendix 1—figure 1* for similar projections for DALYs impact, *Table 3* and S1 for scenario averages over the entire time period, and *Appendix 1—tables 3*, *5*, *7*, and *Appendix 1—table 11* for absolute numbers of deaths).

In the case of measles, Bangladesh initially postponed its campaign by a few months. The two measles models give slightly different predictions about the consequences of this. The Penn State

**Table 1.** a Vaccine impact models – Summary characteristics of the transmission dynamic vaccine impact models for measles (three models).

For IDM, separate information is shown for the models used for Ethiopia and Nigeria.

| Infection | Measles | Measles | Measles | Measles |
|---|---|---|---|---|
| Model name | DynaMICE | IDM (Ethiopia) | IDM (Nigeria) | Penn State |
| Reference | *Verguet et al., 2015* | *Thakkar et al., 2019* | *Zimmermann et al., 2019* | *Chen et al., 2012* |
| Structure | Compartmental | Compartmental | Agent-based | Semi-mechanistic |
| Randomness | Deterministic | Stochastic | Stochastic | Stochastic |
| Time step | Weekly | Semi-monthly | Daily | Annual |
| Age stratification | Yes | No | Yes | Yes |
| Model fitting | Not fitted; uses country-specific $R_o$ (basic reproduction number) for measles from fitted models | Fitted to observed monthly WHO case data (2011–2019) | Fitted to time-series, age-distribution, and spatial correlation between districts in case-based surveillance data. | Fitted to observed annual WHO case data (1980–2017) |
| Validation | Validated through comparisons to the Penn State and/or IDM models in two previous model comparison exercises (*Li et al., 2021*; *WHO, 2019a*). Has also been reviewed by WHO's Immunization and Vaccines Implementation Research Advisory Committee (IVIR-AC) (*WHO, 2019b*) | Validated primarily via forecasting tests in Pakistan and Nigeria. For example, see Figure S10 in *Thakkar et al., 2019*. | Calibrated to reproduce regional time series and age distributions of historical measles incidence as presented in *Zimmermann et al., 2019*. Validated through comparison to the DynaMICE and Penn State models in a previous model comparison exercise (*WHO, 2019a*) | Model and performance of parameter estimation was validated through simulation experiments as described in *Eilertson et al., 2019*. Validated through comparisons to the DynaMICE and/or IDM models in two previous model comparison exercises (*Li et al., 2021*; *WHO, 2019a*). Has also been reviewed by WHO's Immunization and Vaccines Implementation Research Advisory Committee (IVIR-AC) in 2017 and 2019 (*WHO, 2019b*). |
| Case importations | None | None | Random | Random |
| Dose dependency (SIA: supplementary immunisation activities, MCV1: measles 1st dose, MCV2: measles 2nd dose) | SIA doses are weakly dependent of MCV1/2 based on *Portnoy et al., 2018* | MCV2 given only to recipients of MCV1; SIA doses independent of MCV1/2 | MCV2 given only to recipients of MCV1; SIA doses independent of MCV1/2 | |
| Countries modelled | Bangladesh, Chad, Ethiopia, Kenya, Nigeria, South Sudan | Ethiopia | Nigeria | Bangladesh, Chad, Ethiopia, Kenya, Nigeria, South Sudan |

**b. Vaccine impact models – Summary characteristics of the transmission dynamic vaccine impact models for meningococcal A (two models).**

| Infection | MenA | | MenA | |
|---|---|---|---|---|
| Model name | Cambridge | | KP | |
| Reference | *Karachaliou et al., 2015* | | *Jackson et al., 2018* | |
| Structure | Compartmental | | Compartmental | |
| Randomness | Stochastic | | Stochastic | |
| Time step | Daily | | Weekly | |
| Age stratification | Yes | | Yes | |
| Model fitting | Not fitted; calibrated by comparing the predictions to evidence on carriage prevalence by age, disease incidence by age, total annual incidence, seasonality and periodicity | | Fitted to carriage prevalence and disease incidence data for Burkina Faso; calibrated for other regions by comparing seasonality and incidence by age to disease surveillance data | |
| Validation | Peer-review, including by IVIR-AC; two publications *Karachaliou et al., 2015*; *Karachaliou Prasinou et al., 2021*; calibration to observed data (although not formally fitted); | | Peer-review of two publications *Jackson et al., 2018*; *Tartof et al., 2013*; out-of-sample validation on incidence after vaccine introduction in Burkina Faso | |
| Case importations | None | | Infectious people immigrate at a rate of 0.1–1 per million population per week | |

*Table 1 continued on next page*

*Table 1 continued*

**b. Vaccine impact models – Summary characteristics of the transmission dynamic vaccine impact models for meningococcal A (two models).**

| | | |
|---|---|---|
| Dose dependency | Not applicable since 2020 campaigns are targeting population missed by the introductory campaign who are too old for routine immunisation | Campaigns preferentially target unvaccinated persons |
| Countries modelled | Burkina Faso, Chad, Niger, Nigeria | |

**c. Vaccine impact models – Summary characteristics of the transmission dynamic vaccine impact models for yellow fever (two models).**

| | | |
|---|---|---|
| Infection | Yellow fever | Yellow fever |
| Model name | Imperial | Notre Dame |
| Reference | *Gaythorpe et al., 2021b* | *Perkins et al., 2021* |
| Structure | Semi-mechanistic | Semi-mechanistic |
| Randomness | Deterministic | Deterministic |
| Time step | Annual | Annual |
| Age stratification | Yes | Yes |
| Model fitting | Bayesian framework fitted to occurrence and serology data | Bayesian framework fitted to incidence and serology data |
| Validation | Peer-review (two publications Garske et al.; Gaythorpe et al. and EYE strategy); calibration to serological survey data and outbreak occurrence data within Bayesian framework. Compared model structures. | Calibration to serological and case data. Cross-validation of multiple alternative models used to inform the construction of a single ensemble prediction via stacked generalization. |
| Case importations | None | None |
| Dose dependency | Random | Random |
| Countries modelled | Democratic Republic of the Congo, Ghana, Nigeria | |

model predicts that delaying the 2020 campaign increases deaths slightly (by 0.03 per 100,000 over 2020–2030) but this increase is not seen in DynaMICE. Conversely, DynaMICE predicts an increase in deaths of 0.35 per 100,000 over 2020–2030 if routine coverage drops by 50%, but this is not seen in the Penn State model; see *Appendix 1—table 2* for further details. For Ethiopia, a reduction in routine coverage is predicted to lead to outbreaks sooner and increases in overall deaths in all three models (DynaMICE, Penn State models and IDM), while postponing the 2020 campaign only increases deaths in the DynaMICE model. The Ethiopian campaign was eventually reinstated only 3 months later than scheduled. For Kenya, the disruption to routine and campaign immunisation was not predicted to lead to increased risk of outbreaks, due to high coverage of the first dose of measles vaccine and better optimally-timed campaigns in preventing outbreaks during 2020–2030,

**Table 2.** Immunisation scenarios.

Scenarios for disruption of routine immunisation and delay of mass vaccination campaigns due to the COVID-19 pandemic for measles vaccination in six countries, meningococcal A vaccination in four countries, and yellow fever vaccination in three countries. The counterfactual comparative scenario (BAU – business as usual) is no disruption to routine or campaign immunisation.

| Immunisation scenario | Routine immunisation (RI) | Campaign immunisation/ Supplementary immunisation activities (SIAs) |
|---|---|---|
| BAU | No disruption | No disruption |
| Postpone 2020 SIAs - > 2021 | No disruption | Postpone 2020 SIAs to 2021 |
| 50% RI | 50% reduction on RI for 2020 | No disruption |
| 50% RI, postpone 2020 SIAs - > 2021 | 50% reduction on RI for 2020 | Postpone 2020 SIAs to 2021 |

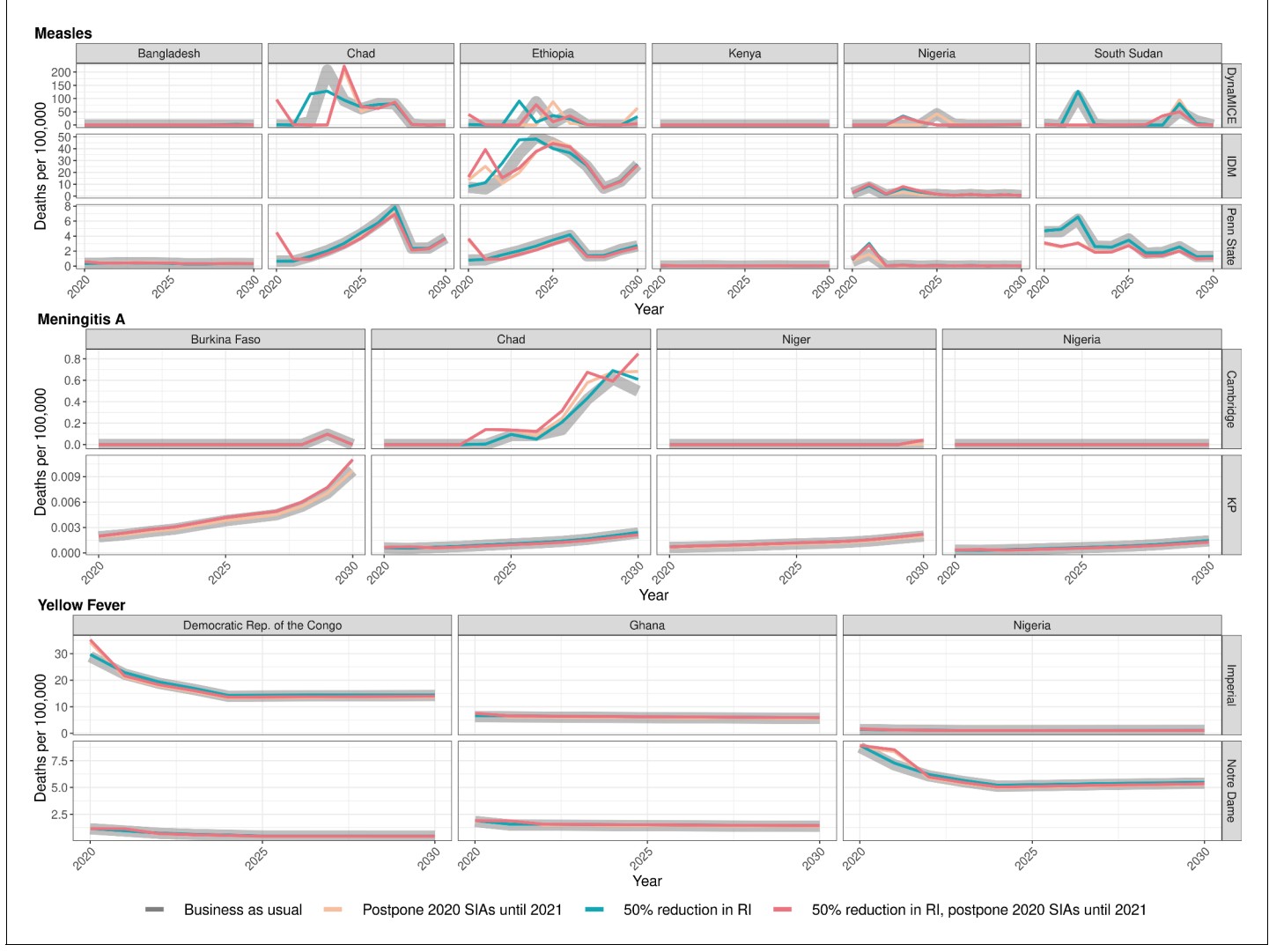

**Figure 1.** Health impact of predicted total deaths for immunisation disruption scenarios and no disruption scenario for measles, meningococcal A, and yellow fever. Model-predicted total deaths per 100,000 population per year for routine immunisation (RI) and campaign immunisation (SIAs – supplementary immunisation activities) disruption scenarios and no disruption scenario (BAU – business-as-usual scenario) for measles, meningococcal A, and yellow fever during 2020–2030.

**Table 3.** Excess deaths per 100,000 between 2020 and 2030 per scenario, infection and modelling group.

Scenarios for disruption of routine immunisation and delay of mass vaccination campaigns due to the COVID-19 pandemic for measles vaccination in six countries, meningococcal A vaccination in four countries, and yellow fever vaccination in three countries. The counterfactual comparative scenario (BAU – business as usual) is no disruption to routine immunisation (RI) or campaign immunisation (SIAs – supplementary immunisation activities). The total of pathogen averages is the sum of the average excess deaths per 100,000 between 2020 and 2030 for each pathogen.

| Scenario | Measles, DynaMICE | Measles, IDM | Measles, Penn State | Men A, Cambridge | Men A, KP | Yellow fever, Imperial | Yellow fever, Notre Dame | Total of pathogen averages |
|---|---|---|---|---|---|---|---|---|
| 50% RI | 1.1569 | 1.1873 | 0.0501 | 0.0020 | 0.0001 | 0.1474 | 0.0755 | 0.9105 |
| Postpone 2020 SIAs -> 2021 | 0.9428 | 0.1248 | −0.0104 | 0.0042 | −0.0001 | −0.0584 | −0.0103 | 0.3202 |
| 50% RI, postpone 2020 SIAs -> 2021 | 0.2401 | 1.3134 | 0.0222 | 0.0064 | 0.0000 | 0.0876 | 0.0536 | 0.5990 |

although coverage of the second-dose of measles vaccine is suboptimal. For Nigeria, either postponement of the 2020 immunisation campaign or a reduction in routine coverage is predicted to lead to increases in measles mortality in Penn State and IDM models, but not in DynaMICE. Note that these increases were predicted to be highly localised in the subnational IDM model; see Discussion for details. For South Sudan, the postponement of immunisation campaigns from 2020 to 2021 is predicted to be beneficial in averting a potential outbreak in 2022 in both DynaMICE and Penn State models (although note caveats in the Discussion about such predictions), but decreases in routine coverage are predicted to lead to more deaths, with a larger predicted increase in DynaMICE. For Chad, both DynaMICE and Penn State models predict an overall increase in deaths with routine coverage drops, but only the Penn State model predicts an increase with a postponement of campaigns. Model-specific estimates of measles deaths per 100,000 over 2020–2030 per country are provided in *Appendix 1—table 2* with absolute numbers for all countries per model given in *Appendix 1—table 11*. Model-specific estimates of measles deaths per 100,000 per year for all countries are provided in *Appendix 1—table 8*.

In the case of meningococcal A (MenA), the short-term disruption to routine immunisation in Burkina Faso, Niger, Nigeria, and Chad, as well as the short-term disruption of immunisation campaigns in Nigeria and Chad would not have a significant impact on the disease incidence (see *Appendix 1—table 4* for model-specific estimates by country). These four countries conducted mass preventive campaigns targeting 1- to 29-year-old populations between 2010 and 2014, and introduced the vaccine into their routine immunization schedules between 2016 and 2019. Niger and Burkina Faso completed catch-up campaigns concomitantly with the introduction into routine, and Chad and Nigeria have started but not completed their catch-up campaigns. A maximum of a 4% increase in MenA deaths over the long term is projected in either of the models and with minimal change in the short term of within 5 years. This is because of the persistence of protection against MenA due to the vaccination strategy combining mass vaccination campaign and routine introduction, which led to a lasting interruption of transmission, in particular from the direct and indirect effects of the initial mass campaigns of the 1- to 29-year-old population in 2010–2014. Model-specific estimates of meningococcal A deaths per 100,000 per year for all countries are provided in *Appendix 1—table 9*.

In the case of yellow fever, for the Democratic Republic of Congo and Nigeria, the postponement of immunisation campaigns from 2020 to 2021 was predicted to cause a short-term increase in burden but when campaigns were implemented, the overall burden was reduced for the time period. A reduction in routine immunization during 2020 was predicted to increase burden over the same period 2020–2030. For Ghana, the postponement of immunisation campaigns from 2020 to 2021 did not lead to an increase in yellow fever burden in the short-term, whereas a reduction in routine immunization in 2020 was predicted to increase the yellow fever burden by 0.33 or 0.07 deaths per 100,000 between 2020 and 2030 in the Imperial and Notre Dame models respectively. Model-specific estimates of excess deaths by country from 2020 to 2030 are shown in *Appendix 1—table 6*. Neither model was designed to specifically capture yellow fever outbreak dynamics. Therefore, although the delay of immunisation campaigns was predicted to reduce the burden of yellow fever for 2020–2030 in select settings by a small (less than 1%) amount, the increased risk of an outbreak is not accounted for in the models and this could outweigh the predicted long-term benefits. Model-specific estimates of yellow fever deaths per 100,000 per year for all countries are provided in *Appendix 1—table 10*.

*Figure 2* shows the model-predicted excess deaths per 100,000 population per year by model for routine and campaign immunisation disruption scenarios in comparison to no disruption scenario for measles, meningococcal A, and yellow fever. The excess deaths are summed over 2020–2030 (see *Appendix 1—figure 2* for similar projections for DALYs impact). The scale of excess mortality due to the immunisation service disruptions are higher for measles vaccination in comparison to meningococcal A and yellow fever vaccination; indeed excess mortality is minimal for meningococcal A.

*Appendix 1—figure 3* shows the normalised model-predicted excess deaths per year and country by model for routine and campaign immunisation disruption scenarios in comparison to the no disruption scenario for measles, meningococcal A, and yellow fever, with excess deaths summed and normalised over 2020–2030 (see *Appendix 1—figure 4* for similar projections for DALYs impact). For measles, there are differences between models but usually a 50% reduction in routine

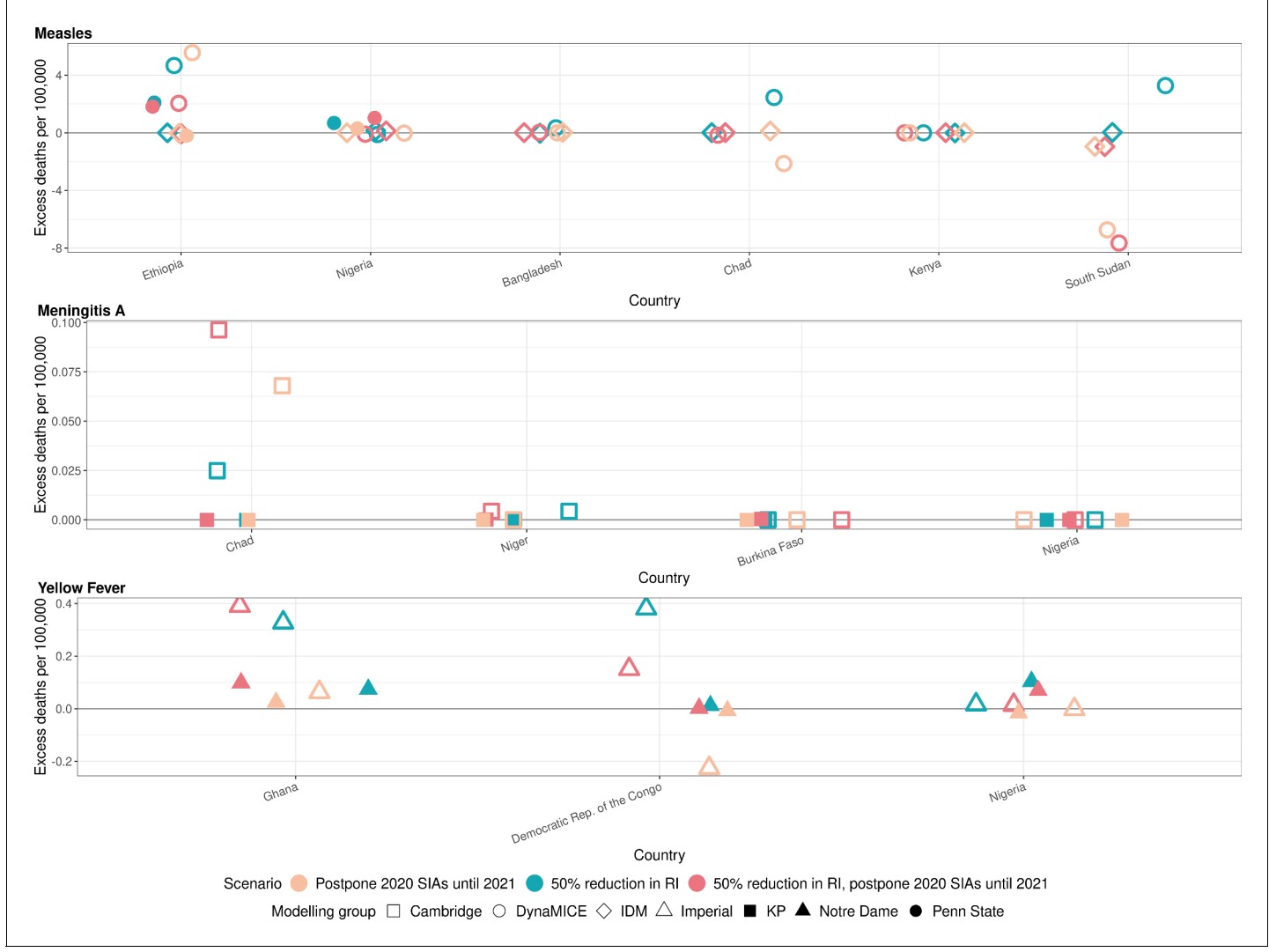

**Figure 2.** Health impact of excess deaths for immunisation disruption scenarios in comparison to no disruption scenario for measles, meningococcal A, and yellow fever. Model-predicted excess deaths per 100,000 population per year for routine immunisation (RI) and campaign immunisation (SIAs – supplementary immunisation activities) disruption scenarios in comparison to no disruption scenario (BAU – business-as-usual scenario) for measles, meningococcal A, and yellow fever. Excess deaths are summed over 2020–2030.

immunisation was projected to increase the excess deaths the most in comparison to scenarios involving the postponement of immunisation campaigns from 2020 to 2021. For MenA, a 50% reduction in routine immunisation and the postponement of immunisation campaigns from 2020 to 2021 was projected to increase the excess deaths the most for Chad, although the scale of absolute impact is minimal (see *Figure 2*). For yellow fever, a reduction in routine immunisation was projected to increase the excess deaths the most (either in conjunction with campaign delay or not). Whilst the postponement of immunisation campaigns from 2020 to 2021 appears to have a beneficial impact of lower deaths in comparison to immunisation campaigns in 2020 for the Democratic Republic of the Congo in *Appendix 1—figure 3*, this does not capture the short-term increase in burden due to the missed campaign. The beneficial effect is due solely to the proportionally larger campaign implemented in 2021, that is a campaign with the same coverage leads to more fully vaccinated persons as the population grows.

# Discussion

The health impact of routine immunisation service disruptions and mass vaccination campaign suspensions due to the COVID-19 pandemic differs widely between infections and countries, so decision-makers need to consider their local epidemiological situation. For meningococcal A and yellow fever, we predict that postponing campaigns has a minimal short-term effect because both pathogens have a low effective reproduction number and strong existing herd immunity from recent campaigns in the countries modelled (see *Appendix 1—table 17* for list of campaigns). However, this is influenced by the model structures and their propensity to capture outbreak dynamics, which particularly affects the predictions for yellow fever. For measles, in some countries such as Ethiopia and Nigeria, even a 1-year postponement of immunisation campaigns could have led to large outbreaks, but both countries were able to implement planned SIAs in 2020 after a few months' delay. In other countries with high routine immunisation coverage and/or recent campaigns, SIAs may be postponed by a year without causing large outbreaks. However, model projections about future outbreaks differ between models in terms of both timing and magnitude. These differences capture uncertainty around data and model structure that differ between models.

In some of our modelled scenarios, postponement of immunisation campaigns does not appear to increase overall cases, if the delay time-period is less than the interval to the next outbreak. Such a scenario is inferred in the immunisation disruption scenarios for postponement of measles campaigns for South Sudan. This does not imply that a postponement is preferred, as we do not take into account other contextual or programmatic factors; rather it reflects the effectiveness of campaigns in closing the immunity gaps and the demographic effect of including more children in delayed campaigns. In instances with very low routine immunisation coverage, there is a possibility that the vaccination campaign is the main opportunity for missed children to be vaccinated. Thus, for the same proportion of the same age group targeted by campaigns, more children will be vaccinated for the same coverage levels in countries with birth rates increasing over time. While these results may be useful in the COVID-19 context, there is also considerable uncertainty around both model findings and data inputs such as incidence and vaccine coverage that prohibits further general comment on the optimal timing of campaigns.

The measles immunisation campaigns for 2020 in Nigeria were specifically targeted at Kogi and Niger states, states that were originally scheduled for inclusion in the campaigns for 2019 across northern Nigeria which were delayed for other reasons. Given the localised build-up of susceptibility in these two states due to low routine immunisation coverage and the long window between campaigns, IDM's subnational Nigeria model indicated that further campaign delays would result in a high risk of localised outbreaks in these states (one potential explanation for the IDM model predicting worse consequences of delays in these campaigns than the other models). Campaigns targeted specifically to these two states were implemented in October 2020. In general, for countries where routine immunisation coverage was low even before the COVID-19 pandemic, the build-up of the susceptible population from low routine immunisation coverage over 2–3 years between campaigns enhances the risk of outbreaks more than recent and temporary disruptions to routine immunisation. Further, our models did not include the possibility that COVID-19 restrictions may have temporarily reduced measles transmissibility and the risk of measles outbreaks due to reduced chance of introduction of infection into populations with immunity gaps. This risk rises again rapidly once travel restrictions and physical distancing are relaxed. This is an additional reason (which we do not model) for implementing postponed immunisation campaigns at the earliest opportunity to prevent measles outbreaks as COVID-19 restrictions are lifted (*LSHTM CMMID COVID-19 Working Group et al., 2021*).

While the degree of health impact of service disruptions varies, the models generally show that reductions in routine immunisation coverage have a far greater impact on predicted excess deaths over the next decade for all infections modeled than postponement of campaigns. This has significant implications for countries planning catch-up strategies and highlights the need for increased emphasis on the importance of implementing catch-up as an ongoing part of routine immunisation (*WHO, 2020f*).

The disease burden averted by measles and meningococcal A vaccination are primarily among under-5-year-old and under-10-year-old children respectively, and disease burden averted by yellow fever vaccination are among younger age-group individuals. Since children and younger age-group

individuals are at relatively lower risk of morbidity and mortality from COVID-19 in comparison to elderly populations, the health benefits of sustaining measles, meningococcal A, and yellow fever immunisation programmes during the COVID-19 pandemic outweigh the excess SARS-CoV-2 infection risk to these age groups that are associated with vaccination service delivery points. Thereby, the delivery of measles, meningococcal A, and yellow fever immunisation services should continue, as logistically as possible, by adapting service delivery in a COVID-secure manner with implementation of SARS-CoV-2 infection prevention and control measures.

Our study has limitations and we have not considered logistical constraints posed by the COVID-19 prevention and control measures on vaccine supply, demand for vaccination, access, and health workforce. Future introduction of COVID-19 vaccination may also divert the workforce normally conducting campaigns for other vaccines. Our models do not reflect geographical heterogeneity subnationally, whereas in reality this is a key feature. Nor do we incorporate known seasonality of infections, which may affect the window of opportunity for catching up. The models used in this analysis, in particular for yellow fever, are best suited to capture long-term changes in disease burden due to vaccination and cannot capture outbreak dynamics that may arise in the short-term. A key strength of our analysis is that we used two to three models for each infection, which allowed investigation of whether projections were sensitive to model structure and assumptions. Each model had different strengths and limitations. For instance, some models measured epidemic properties like reproduction numbers directly, while other models used estimates from other studies. We did indeed find quantitative differences between models of the same infection, but most models agreed on the countries in which disruptions had the largest effect on disease burden.

A further limitation is the omission of changes to transmission in the three pathogens due to COVID-19 mitigation measures. This is a critical area that needs further investigation; however, all three included pathogens have substantially different dynamics to those of SARS-CoV-2. For *yellow fever*, the majority of transmission is sylvatic rather than person-to-person, so COVID-19 mitigation measures are unlikely to have a major effect on incidence, unless they decrease contact between humans and the sylvatic cycle. For *meningococcal A*, we find that even with a decrease in vaccine coverage there is limited potential for outbreaks, so decreased transmission due to COVID-19 non-pharmaceutical interventions will only reinforce this. For *measles*, there is the potential for non-pharmaceutical interventions to decrease transmission. However, measles is much more transmissible than COVID-19 (with $R_0$ usually well above 10 rather than 2–5 *Guerra et al., 2017*), and transmission is generally concentrated among very young children rather than adults. Hence it is unclear whether interventions designed for COVID-19 (mask wearing, closure of schools, workplaces and retail, travel restrictions etc.) will be able to prevent measles outbreaks. Further, while COVID-19 mitigation measures may temporarily reduce measles transmissibility and outbreak risk from measles immunity gaps, the risk for measles outbreaks will rise rapidly once COVID-19-related contact restrictions are lifted (*LSHTM CMMID COVID-19 Working Group et al., 2021*), which happens at different rates in different parts of countries.

We conducted our health impact assessment to align with the WHO framework for decision making using an evidence-based approach to assist in prioritisation of vaccines and strategies for catch-up vaccination during the COVID-19 pandemic (*WHO, 2020f*). The framework highlights three main steps, with the primary step being an epidemiological risk assessment for each disease based on the burden of disease and population immunity, as well as the risk factors associated with the immunisation service disruptions. The second step focuses on the amenability of delivery strategies and operational factors for each vaccine, and the third step on the assessment of contextual factors and competing needs.

Our health impact assessment addresses in part the primary step of an epidemiological risk assessment by estimating the disease burden for different immunisation scenarios, but does not include the health impact assessment of excess COVID-19 disease burden attributable to these immunisation scenarios. While we have assessed the immunity gaps caused by immunisation service disruptions for measles, meningococcal A, and yellow fever vaccination in 10 low- and lower middle-income countries, sustaining routine immunisation and resuming immunisation campaigns during the COVID-19 pandemic requires adaptations to service delivery with additional safety measures to protect the health workers and the community from SARS-CoV-2 infection (*Banks and Boonstoppel, 2020*). Infection prevention and control measures include personal protective equipment for health workers, children to be vaccinated, and their parents or caregivers; additional prevention and control

measures against SARS-CoV-2 infection at vaccination sites; physical distancing; and symptomatic screening and triaging (*WHO, 2020e*). COVID-19 transmission may be further mitigated by delivering several vaccines during a single campaign (such as measles and polio vaccines), or even combining vaccines with other age-relevant interventions such as nutritional supplements. Further, social mobilisation is needed to address the rumours, misinformation, and fear among the community to access vaccination safely during the COVID-19 pandemic (*WHO, 2020g*). Therefore, our health impact assessment needs to be followed up by planning and implementation of catch-up vaccination to close the immunity gaps using a mixture of locally appropriate strategies to strengthen immunisation (*Cutts et al., 2021*), alongside access to additional operating costs to conduct routine and campaign immunisation services safely in COVID-secure environments while considering contextual factors and competing needs.

## Data availability

All code, data inputs and outputs used to generate the results in the manuscript (apart from projections about vaccine coverage beyond 2020 which are commercially confidential property of Gavi) are available at: https://github.com/vimc/vpd-covid-phase-I (*Gaythorpe, 2021a*; copy archived at swh:1: rev:ebff9a24b8b7c9a7c6c5c77f783f2435a57d1d2b).

## Acknowledgements

We would like to thank the following non-author collaborators: Jim Alexander, Laurence Alcyone Cibrelus Yamamoto, Natasha Crowcroft, Heather Ferguson, Neil Ferguson, James Goodson, Brittany Hagedorn, Lee Hampton, Lee Lee Ho, Dan Hogan, Raymond Hutubessy, Sudhir Khanal, Balcha Girma Masresha, Jonathan Mosser, Mark Papania, Bryan Patenaude, William Augusto Perea Caro, Robert Perry, Jeff Pituch, Allison Portnoy, Marie-Pierre Preziosi, Cassandra Quintanilla Angulo, Olivier Ronveaux, Sara Sa Silva, Yodit Sahlemariam, Alyssa Sbarra, Yoonie Sim, David Sniadack, Matthew Steele, Claudia Steulet, Peter Strebel, Aaron Wallace, Susan Wang, Xinhu Wang, Kirsten Ward, Libby Watts, and Karene Yeung. IDM authors would like to acknowledge our colleagues at the National Primary Health Development Agency and NCDC (Ministry of Health of Nigeria), World Health Organization Nigeria office and WHO Headquarters for collecting and providing surveillance data for model development. Funding: This study was carried out as part of the Vaccine Impact Modelling Consortium, and funded by Gavi, the Vaccine Alliance and the Bill and Melinda Gates Foundation (OPP115270 / INV-009125 and INV016832). This publication is based on research funded in part by the Bill and Melinda Gates Foundation, including but not limited to models and data analysis performed by the Institute for Disease Modeling at the Bill and Melinda Gates Foundation. The funders were involved in study design, data collection, analysis and interpretation, report writing, and the decision to submit for publication. All authors had full access to all the data in the study and had final responsibility for the decision to submit for publication. The views expressed are those of the authors and not necessarily those of the Consortium or its funders.

## Additional information

### Group author details

**VIMC Working Group on COVID-19 Impact on Vaccine Preventable Disease**
Andre Arsene Bita Fouda: World Health Organization - Regional Office for Africa, Brazzaville, Democratic Republic of the Congo; Felicity Cutts: London School of Hygiene & Tropical Medicine, London, United Kingdom; Emily Dansereau: Bill & Melinda Gates Foundation, Seattle, United States; Antoine Durupt: World Health Organization, Geneva, Switzerland; Ulla Griffiths: United Nations Children's Fund (UNICEF), New York, United States; Jennifer Horton: World Health Organization, Geneva, Switzerland; L Kendall Krause: Bill & Melinda Gates Foundation, Seattle, United States; Katrina Kretsinger: World Health Organization, Geneva, Switzerland; Tewodaj Mengistu: Gavi, the Vaccine Alliance, Geneva, Switzerland; Imran Mirza: United Nations Children's Fund (UNICEF), New York, United States; Simon R Procter: London School of Hygiene

& Tropical Medicine, London, United Kingdom; Stephanie Shendale: World Health Organization, Geneva, Switzerland

**Competing interests**
Mark Jit: Reviewing editor, eLife. Niket Thakkar, Kevin McCarthy: NT and KM are employees of the Institute for Disease Modeling at the Bill and Melinda Gates Foundation. This publication is based on research funded in part by the Bill and Melinda Gates Foundation, including but not limited to models and data analysis performed by the Institute for Disease Modeling at the Bill and Melinda Gates Foundation. VIMC Working Group on COVID-19 Impact on Vaccine Preventable Disease: FC declares consultancy fees from the Bill and Melinda Gates Foundation. TM is an employee of Gavi, which funded the research. ED and LKK are employees of the Bill and Melinda Gates Foundation, which funded the research. Caroline Trotter: CT declares a consultancy fee from GSK in 2018 (unrelated to the submitted work). The other authors declare that no competing interests exist.

**Funding**

| Funder | Grant reference number | Author |
|---|---|---|
| Gavi, the Vaccine Alliance and the Bill & Melinda Gates Foundation | OPP115270 / INV-009125 and INV016832 | Katy A M Gaythorpe<br>Kaja Abbas<br>John Huber<br>Andromachi Karachaliou<br>Niket Thakkar<br>Kim Woodruff<br>Xiang Li<br>Susy Echeverria-Londono<br>Matthew Ferrari<br>Michael L Jackson<br>Kevin McCarthy<br>T Alex Perkins<br>Caroline Trotter<br>Mark Jit |

The funders were involved in study design, data collection, analysis and interpretation, report writing, and the decision to submit for publication. All authors had full access to all the data in the study and had final responsibility for the decision to submit for publication. The views expressed are those of the authors and not necessarily those of the Consortium or its funders.

**Author contributions**
Katy AM Gaythorpe, Niket Thakkar, Kaja Abbas, John Huber, Andromachi Karachaliou, Kim Woodruff, Xiang Li, Susy Echeverria-Londono, Matthew Ferrari, Michael L Jackson, Kevin McCarthy, T Alex Perkins, Caroline Trotter, Mark Jit, Conceptualization, Data curation, Formal analysis, Validation, Investigation, Visualization, Methodology, Writing - original draft, Project administration, Writing - review and editing; VIMC Working Group on COVID-19 Impact on Vaccine Preventable Disease, Conceptualization, Data curation, Visualization, Writing - review and editing

**Author ORCIDs**
Katy AM Gaythorpe [ID] https://orcid.org/0000-0003-3734-9081
John Huber [ID] http://orcid.org/0000-0001-5245-5187
Kim Woodruff [ID] https://orcid.org/0000-0003-4618-8267
T Alex Perkins [ID] https://orcid.org/0000-0002-7518-4014
Mark Jit [ID] https://orcid.org/0000-0001-6658-8255

**Decision letter and Author response**
Decision letter https://doi.org/10.7554/eLife.67023.sa1
Author response https://doi.org/10.7554/eLife.67023.sa2

## Additional files

### Supplementary files

• Transparent reporting form

### Data availability

All code, data inputs and outputs used to generate the results in the manuscript (apart from projections about vaccine coverage beyond 2020 which are commercially confidential property of Gavi) are available at: https://github.com/vimc/vpd-covid-phase-I (copy archived at https://archive.software-heritage.org/swh:1:rev:ebff9a24b8b7c9a7c6c5c77f783f2435a57d1d2b).

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

## Appendix 1

## Section 1. Tables (appendix)

**Appendix 1—table 1.** Excess disability-adjusted life years (DALYs) per 100,000 between 2020 and 2030 per scenario, infection and modelling group.

Scenarios for disruption of routine immunisation (RI) and delay of mass vaccination campaigns (SIAs – supplementary immunisation activities) due to the COVID-19 pandemic for measles vaccination in six countries, meningococcal A vaccination in four countries, and yellow fever vaccination in three countries. The counterfactual comparative scenario (BAU – business as usual) is no disruption to routine or campaign immunisation.

| Scenario | Measles, DynaMICE | Measles, IDM | Measles, Penn State | Men A, Cambridge | Men A, KP | Yellow fever, Imperial | Yellow fever, Notre Dame |
|---|---|---|---|---|---|---|---|
| 50% RI | 79.2110 | 68.5537 | 2.7503 | 0.1175 | 0.0037 | 9.3283 | 4.3831 |
| Postpone 2020 SIAs - > 2021 | 69.9709 | 5.7308 | −0.0990 | 0.2650 | −0.0027 | −2.7355 | −0.5797 |
| 50% RI, postpone 2020 SIAs - > 2021 | 17.0570 | 74.0683 | 1.6898 | 0.4017 | 0.0004 | 6.5284 | 3.1370 |

**Appendix 1—table 2.** Excess measles deaths per 100,000 between 2020 and 2030 per scenario, country and modelling group.

The counterfactual comparative scenario (BAU – business as usual) is no disruption to routine immunisation (RI) or campaign immunisation (SIAs – supplementary immunisation activities). Countries shown are Bangladesh (BGD), Ethiopia (ETH), Kenya (KEN), Nigeria (NGA), South Sudan (SSD), and Chad (TCD).

| Country | 50% RI, DynaMICE | 50% RI, IDM | 50% RI, Penn State | Postpone 2020 SIAs - > 2021, DynaMICE | Postpone 2020 SIAs - > 2021, IDM | Postpone 2020 SIAs - > 2021, Penn State | 50% RI, postpone 2020 SIAs - > 2021, DynaMICE | 50% RI, postpone 2020 SIAs - > 2021, IDM | 50% RI, postpone 2020 SIAs - > 2021, Penn State |
|---|---|---|---|---|---|---|---|---|---|
| BGD | 0.35 | NA | −0.03 | 0 | NA | 0.03 | 0.03 | NA | 0.01 |
| ETH | 4.67 | 2.1 | 0 | 5.56 | −0.19 | −0.03 | 2.05 | 1.82 | −0.07 |
| KEN | 0 | NA | −0.01 | 0 | NA | 0.01 | 0 | NA | 0.01 |
| NGA | −0.12 | 0.68 | 0.15 | −0.02 | 0.3 | 0.01 | −0.09 | 1.03 | 0.13 |
| SSD | 3.28 | NA | 0.03 | −6.73 | NA | −0.95 | −7.65 | NA | −0.96 |
| TCD | 2.45 | NA | 0.02 | −2.13 | NA | 0.12 | −0.16 | NA | 0.01 |

**Appendix 1—table 3.** Excess measles deaths between 2020 and 2030 per scenario, country and modelling group.

The counterfactual comparative scenario (BAU – business as usual) is no disruption to routine immunisation (RI) or campaign immunisation (SIAs – supplementary immunisation activities). Countries shown are Bangladesh (BGD), Ethiopia (ETH), Kenya (KEN), Nigeria (NGA), South Sudan (SSD), and Chad (TCD).

| Country | 50% RI, DynaMICE | 50% RI, IDM | 50% RI, Penn State | Postpone 2020 SIAs - > 2021, DynaMICE | Postpone 2020 SIAs - > 2021, IDM | Postpone 2020 SIAs - > 2021, Penn State | 50% RI, postpone 2020 SIAs - > 2021, DynaMICE | 50% RI, postpone 2020 SIAs - > 2021, IDM | 50% RI, postpone 2020 SIAs - > 2021, Penn State |
|---|---|---|---|---|---|---|---|---|---|
| BGD | 6552 | NA | −539 | 0 | NA | 593 | 578 | NA | 276 |

*Continued on next page*

*Appendix 1—table 3 continued*

| Country | 50% RI, DynaMICE | 50% RI, IDM | 50% RI, Penn State | Postpone 2020 SIAs -> 2021, DynaMICE | Postpone 2020 SIAs -> 2021, IDM | Postpone 2020 SIAs -> 2021, Penn State | 50% RI, postpone 2020 SIAs -> 2021, DynaMICE | 50% RI, postpone 2020 SIAs -> 2021, IDM | 50% RI, postpone 2020 SIAs -> 2021, Penn State |
|---|---|---|---|---|---|---|---|---|---|
| ETH | 66678 | 29951 | 40 | 79384 | −2783 | −473 | 29241 | 25981 | −946 |
| KEN | 0 | NA | −40 | 0 | NA | 64 | 0 | NA | 59 |
| NGA | −3016 | 17545 | 3919 | −634 | 7777 | 137 | −2430 | 26559 | 3427 |
| SSD | 4493 | NA | 44 | −9229 | NA | −1298 | −10485 | NA | −1316 |
| TCD | 5125 | NA | 35 | −4460 | NA | 260 | −333 | NA | 29 |

**Appendix 1—table 4.** Excess meningococcal A deaths per 100,000 between 2020 and 2030 per scenario, country and modelling group.
The counterfactual comparative scenario (BAU – business as usual) is no disruption to routine immunisation (RI) or campaign immunisation (SIAs – supplementary immunisation activities). Countries shown are Burkina Faso (BFA), Niger (NER), Nigeria (NGA), and Chad (TCD).

| Country | 50% RI, Cambridge | 50% RI, KP | Postpone 2020 SIAs -> 2021, Cambridge | Postpone 2020 SIAs -> 2021, KP | 50% RI, postpone 2020 SIAs -> 2021, Cambridge | 50% RI, postpone 2020 SIAs -> 2021, KP |
|---|---|---|---|---|---|---|
| BFA | 0 | 0 | 0 | 0 | 0 | 0 |
| NER | 0 | 0 | 0 | 0 | 0 | 0 |
| NGA | 0 | 0 | 0 | 0 | 0 | 0 |
| TCD | 0.02 | 0 | 0.07 | 0 | 0.1 | 0 |

**Appendix 1—table 5.** Excess meningococcal A deaths between 2020 and 2030 per scenario, country and modelling group.
The counterfactual comparative scenario (BAU – business as usual) is no disruption to routine immunisation (RI) or campaign immunisation (SIAs – supplementary immunisation activities). Countries shown are Burkina Faso (BFA), Niger (NER), Nigeria (NGA), and Chad (TCD).

| Country | 50% RI, Cambridge | 50% RI, KP | Postpone 2020 SIAs -> 2021, Cambridge | Postpone 2020 SIAs -> 2021, KP | 50% RI, postpone 2020 SIAs -> 2021, Cambridge | 50% RI, postpone 2020 SIAs -> 2021, KP |
|---|---|---|---|---|---|---|
| BFA | 0 | 1 | 0 | 0 | 0 | 1 |
| NER | 14 | 0 | 0 | 0 | 14 | 0 |
| NGA | 0 | 0 | 0 | -2 | 0 | -1 |
| TCD | 52 | 0 | 142 | 0 | 201 | 0 |

**Appendix 1—table 6.** Excess yellow fever deaths per 100,000 between 2020 and 2030 per scenario, country and modelling group.
The counterfactual comparative scenario (BAU – business as usual) is no disruption to routine immunisation (RI) or campaign immunisation (SIAs – supplementary immunisation activities). Countries shown are the Democratic Republic of Congo (COD), Ghana (GHA) and Nigeria (NGA).

| Country | 50% RI, Imperial | 50% RI, Notre Dame | Postpone 2020 SIAs -> 2021, Imperial | Postpone 2020 SIAs -> 2021, Notre Dame | 50% RI, postpone 2020 SIAs -> 2021, Imperial | 50% RI, postpone 2020 SIAs -> 2021, Notre Dame |
|---|---|---|---|---|---|---|
| COD | 0.38 | 0.01 | −0.23 | −0.01 | 0.15 | 0 |

*Continued on next page*

*Appendix 1—table 6 continued*

| Country | 50% RI, Imperial | 50% RI, Notre Dame | Postpone 2020 SIAs - > 2021, Imperial | Postpone 2020 SIAs - > 2021, Notre Dame | 50% RI, postpone 2020 SIAs - > 2021, Imperial | 50% RI, postpone 2020 SIAs - > 2021, Notre Dame |
|---|---|---|---|---|---|---|
| GHA | 0.33 | 0.07 | 0.06 | 0.02 | 0.39 | 0.1 |
| NGA | 0.02 | 0.1 | 0 | −0.02 | 0.01 | 0.07 |

**Appendix 1—table 7.** Excess yellow fever deaths between 2020 and 2030 per scenario, country and modelling group.
The counterfactual comparative scenario (BAU – business as usual) is no disruption to routine immunisation (RI) or campaign immunisation (SIAs – supplementary immunisation activities). Countries shown are the Democratic Republic of Congo (COD), Ghana (GHA), and Nigeria (NGA).

| Country | 50% RI, Imperial | 50% RI, Notre Dame | Postpone 2020 SIAs - > 2021, Imperial | Postpone 2020 SIAs - > 2021, Notre Dame | 50% RI, postpone 2020 SIAs - > 2021, Imperial | 50% RI, postpone 2020 SIAs - > 2021, Notre Dame |
|---|---|---|---|---|---|---|
| COD | 4379 | 137 | −2590 | −88 | 1731 | 25 |
| GHA | 1241 | 281 | 239 | 94 | 1481 | 375 |
| NGA | 421 | 2675 | −45 | −426 | 377 | 1798 |

**Appendix 1—table 8.** Excess measles deaths per 100,000 per year between 2020 and 2030 per scenario, year and modelling group.
The counterfactual comparative scenario (BAU – business as usual) is no disruption to routine immunisation (RI) or campaign immunisation (SIAs – supplementary immunisation activities).

| Year | 50% RI, DynaMICE | 50% RI, IDM | 50% RI, Penn State | Postpone 2020 SIAs - > 2021, DynaMICE | Postpone 2020 SIAs - > 2021, IDM | Postpone 2020 SIAs - > 2021, Penn State | 50% RI, postpone 2020 SIAs - > 2021, DynaMICE | 50% RI, postpone 2020 SIAs - > 2021, IDM | 50% RI, postpone 2020 SIAs - > 2021, Penn State |
|---|---|---|---|---|---|---|---|---|---|
| 2020 | 0 | 0.34 | 0.06 | 10.46 | 2.32 | 0.7 | 10.46 | 3.25 | 0.82 |
| 2021 | 0 | 2.7 | 0.55 | 0.19 | 8.32 | 0.02 | 0.2 | 13.55 | 0.44 |
| 2022 | 3.44 | 4.98 | 0 | −2.52 | −1.27 | −0.19 | −2.52 | 0.77 | −0.2 |
| 2023 | 28.56 | 6.57 | 0 | −6.31 | −5.48 | −0.1 | 4.96 | −0.96 | −0.12 |
| 2024 | −11.77 | 0.72 | −0.03 | −14.68 | −4.5 | −0.12 | 5.96 | −2.48 | −0.16 |
| 2025 | −9.82 | −1.52 | 0.02 | 16.07 | 0.67 | −0.1 | −14.65 | 0.03 | −0.13 |
| 2026 | −5.38 | −0.26 | 0.01 | −7.57 | 1.83 | −0.1 | −3.38 | 1.51 | −0.11 |
| 2027 | 0.24 | 0.23 | 0.02 | 0.37 | 0.36 | −0.03 | 1.08 | 0.6 | −0.04 |
| 2028 | 0.79 | 0.08 | 0 | 0.88 | −0.02 | −0.03 | −0.02 | 0.01 | −0.05 |
| 2029 | 0.55 | 0.09 | −0.02 | −0.3 | −0.03 | −0.05 | 0.15 | −0.16 | −0.06 |
| 2030 | 6.55 | 0.09 | 0 | 12.95 | −0.26 | −0.04 | 1.45 | −0.1 | −0.05 |

**Appendix 1—table 9.** Excess meningococcal A deaths per 100,000 per year between 2020 and 2030 per scenario, year and modelling group.
The counterfactual comparative scenario (BAU – business as usual) is no disruption to routine immunisation (RI) or campaign immunisation (SIAs – supplementary immunisation activities).

| Year | 50% RI, Cambridge | 50% RI, KP | Postpone 2020 SIAs -> 2021, Cambridge | Postpone 2020 SIAs -> 2021, KP | 50% RI, postpone 2020 SIAs -> 2021, Cambridge | 50% RI, postpone 2020 SIAs -> 2021, KP |
|---|---|---|---|---|---|---|
| 2020 | 0 | 0 | 0 | 0 | 0 | |
| 2021 | 0 | 0 | 0 | 0 | 0 | 0 |
| 2022 | 0 | 0 | 0 | 0 | 0 | 0 |
| 2023 | 0 | 0 | 0 | 0 | 0 | 0 |
| 2024 | 0 | 0 | 0.01 | 0 | 0.01 | 0 |
| 2025 | 0 | 0 | 0 | 0 | 0 | 0 |
| 2026 | 0 | 0 | 0 | 0 | 0 | 0 |
| 2027 | 0 | 0 | 0.01 | 0 | 0.01 | 0 |
| 2028 | 0 | 0 | 0.01 | 0 | 0.02 | 0 |
| 2029 | 0 | 0 | 0 | 0 | 0 | 0 |
| 2030 | 0.01 | 0 | 0.01 | 0 | 0.03 | 0 |

**Appendix 1—table 10.** Excess yellow fever deaths per 100,000 per year between 2020 and 2030 per scenario, year and modelling group.

The counterfactual comparative scenario (BAU – business as usual) is no disruption to routine immunisation (RI) or campaign immunisation (SIAs – supplementary immunisation activities).

| Year | 50% RI, Imperial | 50% RI, Notre Dame | Postpone 2020 SIAs -> 2021, Imperial | Postpone 2020 SIAs -> 2021, Notre Dame | 50% RI, postpone 2020 SIAs -> 2021, Imperial | 50% RI, postpone 2020 SIAs -> 2021, Notre Dame |
|---|---|---|---|---|---|---|
| 2020 | 0.28 | 0.12 | 1.7 | 0 | 2.02 | 0.12 |
| 2021 | 0.22 | 0.12 | −0.36 | 0.86 | −0.15 | 0.98 |
| 2022 | 0.18 | 0.09 | −0.29 | −0.14 | −0.12 | −0.07 |
| 2023 | 0.16 | 0.08 | −0.25 | −0.12 | −0.1 | 0.05 |
| 2024 | 0.13 | 0.07 | −0.2 | −0.1 | −0.07 | 0.04 |
| 2025 | 0.13 | 0.07 | −0.19 | −0.09 | −0.07 | −0.04 |
| 2026 | 0.12 | 0.06 | −0.18 | −0.09 | −0.07 | −0.04 |
| 2027 | 0.12 | 0.06 | −0.18 | −0.09 | −0.06 | −0.04 |
| 2028 | 0.11 | 0.06 | −0.17 | −0.09 | −0.06 | −0.04 |
| 2029 | 0.11 | 0.06 | −0.16 | −0.08 | −0.06 | −0.04 |
| 2030 | 0.11 | 0.06 | −0.16 | −0.08 | −0.06 | −0.04 |

**Appendix 1—table 11.** Excess deaths between 2020 and 2030 per scenario, infection and modelling group.

Scenarios for disruption of routine immunisation and delay of mass vaccination campaigns due to the COVID-19 pandemic for measles vaccination in six countries, meningococcal A vaccination in four countries, and yellow fever vaccination in three countries. The counterfactual comparative scenario (BAU – business as usual) is no disruption to routine immunisation (RI) or campaign immunisation (SIAs – supplementary immunisation activities).

| Scenario | Measles, DynaMICE | Measles, IDM[*] | Measles, Penn State | Men A, Cambridge | Men A, KP | Yellow fever, Imperial | Yellow fever, Notre Dame | Total of pathogen averages[#] |
|---|---|---|---|---|---|---|---|---|
| 50% RI | 79832.13 | 47495.71 | 3459.278 | 66 | 2.174381 | 6042.15 | 3093.147 | 68265.66 |

*Continued on next page*

*Appendix 1—table 11 continued*

| Scenario | Measles, DynaMICE | Measles, IDM* | Measles, Penn State | Men A, Cambridge | Men A, KP | Yellow fever, Imperial | Yellow fever, Notre Dame | Total of pathogen averages# |
|---|---|---|---|---|---|---|---|---|
| Postpone 2020 SIAs - > 2021 | 65061.76 | 4994.18 | −715.324 | 142 | −1.78694 | −2395.67 | −420.914 | 33689.78 |
| 50% RI, postpone 2020 SIAs - > 2021 | 16570.58 | 52540.42 | 1529.764 | 215 | 0.06521 | 3589.54 | 2197.85 | 37556.73 |

* Measles IDM covers only two countries.

# Total of pathogen averages exclude Measles IDM as this covers only two countries.

**Appendix 1—table 12.** Percentage differences in deaths from baseline between 2020 and 2030 per scenario.

Scenarios for disruption of routine immunisation and delay of mass vaccination campaigns due to the COVID-19 pandemic for measles vaccination in six countries, meningococcal A vaccination in four countries, and yellow fever vaccination in three countries. The counterfactual comparative scenario (BAU – business as usual) is no disruption to routine immunisation (RI) or campaign immunisation (SIAs – supplementary immunisation activities).

| Scenario | Percentage difference from baseline |
|---|---|
| 50% RI | 9.885481 |
| Postpone 2020 SIAs - > 2021 | 3.780423 |
| 50% RI, postpone 2020 SIAs - > 2021 | 4.802334 |

**Appendix 1—table 13.** Percentage differences in deaths from baseline between 2020 and 2030 per scenario, infection and modelling group.

Scenarios for disruption of routine immunisation and delay of mass vaccination campaigns due to the COVID-19 pandemic for measles vaccination in six countries, meningococcal A vaccination in four countries, and yellow fever vaccination in three countries. The counterfactual comparative scenario (BAU – business as usual) is no disruption to routine immunisation (RI) or campaign immunisation (SIAs – supplementary immunisation activities).

| Scenario | Measles, DynaMICE | Measles, IDM | Measles, Penn State | Men A, Cambridge | Men A, KP | Yellow fever, Imperial | Yellow fever, Notre Dame |
|---|---|---|---|---|---|---|---|
| 50% RI | 19.1957 | 11.9708 | 6.4030 | 15.9806 | 5.8875 | 2.5026 | 1.9177 |
| Postpone 2020 SIAs - > 2021 | 15.6442 | 1.2587 | −1.3240 | 34.3826 | −4.8384 | −0.9923 | −0.2610 |
| 50% RI, postpone 2020 SIAs - > 2021 | 3.9844 | 13.2423 | 2.8315 | 52.0581 | 0.1766 | 1.4868 | 1.3626 |

**Appendix 1—table 14.** Coverage assumptions for the counterfactual comparative scenario (BAU – business as usual), determined through consultation with disease and immunisation programme experts across partners at the global, regional, and national levels.

| Assumption | Measles MCV1: 1 st dose measles vaccine, MCV2: 2nd dose measles vaccine | Yellow fever (YF) | Meningococcal A (Men A) (For countries that have introduced routine) |
|---|---|---|---|

*Continued on next page*

*Appendix 1—table 14 continued*

| Assumption | Measles<br>MCV1: 1 st dose measles vaccine, MCV2: 2nd dose measles vaccine | Yellow fever (YF) | Meningococcal A (Men A) (For countries that have introduced routine) |
|---|---|---|---|
| Routine coverage 2020–2030 (historical coverage from WUENIC – WHO and UNICEF Estimates of National Immunization Coverage) | MCV1: Mean of 2015–2019 coverage MCV2: Highest coverage in 2015–2019 If no MCV2 coverage in 2015–19, assume 50% of MCV1 mean coverage for 2015–19 | YF: Mean of 2015–2019 coverage If no YF coverage in 2015–19, use MCV1 mean coverage for 2015–19 | MenA: Highest coverage in 2015–2019. If no coverage available (for 1 + full years), use MCV1 mean coverage for 2015–19 Exception: where Men A intro age is ≥ 15 m, use MCV2 highest coverage in 2015–19 |
| Vaccine introductions | Assume all countries introduce MCV2 in 2022 if they have not already | Assume all countries introduce YF in 2022 if they have not already | N/A |
| Campaign frequency | Use historic frequency: interval between last two prospectively planned national SIAs (supplementary immunisation activities) | 2019 and 2020 completed and planned campaigns (both planned and reactive) 2021–2030: Mass preventive campaigns as recommended by the WHO EYE strategy (2016), with updated sequencing; no reactive campaigns | 2019 and 2020 completed and planned campaigns 2021–2030: Assume no campaigns |
| Campaign coverage | Use coverage of last national SIA | Assume 85% coverage of the subnational target population for all future campaigns in 2020–2030 (and for 2019 campaigns if actual coverage unavailable). | 2019 and 2020 actual/ forecast campaign coverage level |

**Appendix 1—table 15.** Glossary of terms.

| Term | Description |
|---|---|
| Country | BFA: Burkina Faso<br>BGD: Bangladesh<br>COD: Democratic Republic of the Congo (DRC)<br>ETH: Ethiopia<br>GHA: Ghana<br>KEN: Kenya<br>NER: Niger<br>NGA: Nigeria<br>SSD: South Sudan<br>TCD: Chad |
| Vaccine | MCV1: 1 st dose measles vaccine,<br>MCV2: 2nd dose measles vaccine,<br>YF: yellow fever vaccine,<br>MenA: meningococcal A vaccine |
| Year | Year of vaccination |
| Age from | Minimum age (in years) of the target population |
| Age to | Maximum age (in years) of the target population |
| Age range verbatim | Age of the target population, as provided by WHO or other coverage source |
| Coverage (national level) | Percentage of the target population vaccinated, specified at a national level. |
| Target (national level) | Number of people in the target age range, in the entire country. |

*Continued on next page*

*Appendix 1—table 15 continued*

| Term | Description |
|---|---|
| Subnational campaign | Campaigns which took place sub-nationally, rather than across the whole country. |
| Number vaccinated | Number of individuals vaccinated in a campaign. Where necessary, a demographic cap was applied to constrain the number vaccinated to be no higher than UNWPP records of the total number in the target age group. (UNWPP: United Nations World Population Prospects, 2019 Revision). |
| Affected by COVID-19 | Values are shown for 2020 campaigns only. FALSE: 2020 campaigns unaffected by COVID-19, for example campaigns which took place in early 2020. These campaigns are retained in all disruption scenarios. |

**Appendix 1—table 16.** Routine coverage values used for the counterfactual comparative (business-as-usual) scenario, following the assumptions in **Appendix 1—table 14**.

Target population taken from United Nations World Population Prospects (UNWPP) 2019 revision. Countries: Burkina Faso (BFA), Bangladesh (BGD), Democratic Republic of the Congo (COD), Ethiopia (ETH), Ghana (GHA), Kenya (KEN), Niger (NER), Nigeria (NGA), South Sudan (SSD), Chad (TCD). Vaccines: 1st dose measles vaccine (MCV1), 2nd dose measles vaccine (MCV2), yellow fever vaccine (YF), meningococcal A vaccine (MenA).

| Country | Vaccine | Year | Age from | Age to | Coverage (national level) |
|---|---|---|---|---|---|
| BFA | MenA | 2020–2030 | 0 | 0 | 85% |
| BGD | MCV1 | 2020–2030 | 0 | 0 | 97% |
| | MCV2 | 2020–2030 | 2 | 2 | 93% |
| COD | YF | 2020–2030 | 0 | 0 | 74% |
| ETH | MCV1 | 2020–2030 | 0 | 0 | 64% |
| | MCV2 | 2020–2030 | 2 | 2 | 31% |
| GHA | YF | 2020–2030 | 0 | 0 | 89% |
| KEN | MCV1 | 2020–2030 | 0 | 0 | 92% |
| | MCV2 | 2020–2030 | 2 | 2 | 45% |
| NER | MenA | 2020–2030 | 0 | 0 | 96% |
| NGA | MCV1 | 2020–2030 | 0 | 0 | 61% |
| | MCV2 | 2020–2030 | 2 | 2 | 19% |
| | MenA | 2020–2030 | 0 | 0 | 61% |
| | YF | 2020–2030 | 0 | 0 | 60% |
| SSD | MCV1 | 2020–2030 | 0 | 0 | 51% |
| TCD | MCV1 | 2020–2030 | 0 | 0 | 39% |
| TCD | MenA | 2020–2030 | 0 | 0 | 70% |

**Appendix 1—table 17.** Campaign coverage values used for the counterfactual comparative (business-as-usual) scenario, following the assumptions in **Appendix 1—table 14**.

Countries: Bangladesh (BGD), Democratic Republic of the Congo (COD), Ethiopia (ETH), Ghana (GHA), Kenya (KEN), Nigeria (NGA), South Sudan (SSD), Chad (TCD).

| Country | Vaccine | Year | Age_from | Age_to | Age range verbatim | Coverage (national level) | Target (national level) | Subnational campaign | Number vaccinated | Affected by covid-19 |
|---|---|---|---|---|---|---|---|---|---|---|
| BGD | Measles | 2020 | 1 | 9 | 6M-9Y | 1% | 26,123,496 | yes | 292,437 | FALSE |
| BGD | Measles | 2020 | 1 | 9 | 9M-9Y | 100% | 26,123,496 | no | 26,123,496 | |
| BGD | Measles | 2026 | 1 | 4 | | 93% | 10,972,070 | no | 10,204,025 | |
| COD | YF | 2020 | 1 | 60 | 9M-60Y | 10% | 82,362,957 | yes | 8,468,874 | |
| COD | YF | 2020 | 1 | 60 | 9M-60Y | 8% | 82,362,957 | yes | 6,707,043 | |
| COD | YF | 2021 | 1 | 60 | 9M-60Y | 25% | 84,982,979 | yes | 21,179,612 | |
| COD | YF | 2022 | 1 | 60 | 9M-60Y | 17% | 87,641,611 | yes | 14,875,225 | |
| COD | YF | 2023 | 1 | 60 | 9M-60Y | 14% | 90,340,189 | yes | 12,357,393 | |
| COD | YF | 2024 | 1 | 60 | 9M-60Y | 18% | 93,082,143 | yes | 17,200,562 | |
| ETH | Measles | 2019 | 1 | 14 | 6 M-59M; 6M-14Y | 3% | 41,766,446 | yes | 1,230,934 | |
| ETH | Measles | 2020 | 1 | 4 | 6–59 M | 100% | 13,314,425 | no | 13,314,425 | |
| ETH | Measles | 2027 | 1 | 4 | | 93% | 14,462,250 | no | 13,449,892 | |
| GHA | YF | 2020 | 10 | 60 | 10-60Y | 22% | 21,527,602 | yes | 4,758,966 | |
| KEN | Measles | 2020 | 1 | 4 | 9–59 M | 100% | 5,625,900 | no | 5,625,900 | |
| KEN | Measles | 2024 | 1 | 4 | | 95% | 5,839,639 | no | 5,547,657 | |
| KEN | Measles | 2028 | 1 | 4 | | 95% | 6,220,262 | no | 5,909,249 | |
| NGA | Measles | 2019 | 1 | 9 | 6M-9Y | 1% | 55,695,418 | yes | 436,031 | |
| NGA | Measles | 2019 | 1 | 5 | 6 M-71M | 2% | 32,616,304 | yes | 718,665 | |
| NGA | Measles | 2019 | 1 | 4 | 9–59 M | 81% | 26,413,460 | yes | 21,352,326 | |
| NGA | MenA | 2019 | 1 | 7 | | 55% | 44,499,793 | yes | 24,274,987 | |
| NGA | YF | 2019 | 1 | 44 | 9M-44Y | 0.30% | 167,255,829 | yes | 525,691 | |
| NGA | YF | 2019 | 1 | 44 | 9M-44Y | 1% | 167,255,829 | yes | 1,392,489 | |
| NGA | YF | 2019 | 1 | 44 | 9M-44Y | 1% | 167,255,829 | yes | 1,766,338 | |
| NGA | YF | 2019 | 1 | 44 | 9M-44Y | 4% | 167,255,829 | yes | 6,755,396 | |
| NGA | Measles | 2020 | 1 | 4 | 6–59 M | 7% | 26,844,855 | yes | 1,988,885 | |
| NGA | MenA | 2020 | 7 | 10 | 7–8/9–10 years | 24% | 22,936,865 | yes | 5,618,292 | |
| NGA | MenA | 2020 | 1 | 7 | 1–7 Y | 15% | 45,289,678 | yes | 6,791,329 | |
| NGA | YF | 2020 | 1 | 44 | 9M-44Y | 5% | 171,465,804 | yes | 8,624,060 | FALSE |
| NGA | YF | 2020 | 1 | 44 | 9M-44Y | 3% | 171,465,804 | yes | 4,936,871 | |
| NGA | YF | 2020 | 1 | 44 | 9M-44Y | 16% | 171,465,804 | yes | 26,676,939 | |
| NGA | YF | 2021 | 1 | 44 | 9M-44Y | 20% | 175,731,488 | yes | 34,701,457 | |
| NGA | Measles | 2022 | 1 | 4 | | 88% | 27,691,758 | no | 24,230,288 | |
| NGA | YF | 2022 | 1 | 44 | 9M-44Y | 13% | 180,026,007 | yes | 23,699,548 | |
| NGA | YF | 2023 | 1 | 44 | 9M-44Y | 13% | 184,355,854 | yes | 23,699,548 | |
| NGA | Measles | 2024 | 1 | 4 | | 88% | 28,580,680 | no | 25,008,095 | |
| NGA | Measles | 2026 | 1 | 4 | | 88% | 29,575,232 | no | 25,878,328 | |
| NGA | Measles | 2028 | 1 | 4 | | 88% | 30,532,880 | no | 26,716,270 | |
| NGA | Measles | 2030 | 1 | 4 | | 88% | 31,488,385 | no | 27,552,337 | |
| SSD | Measles | 2020 | 1 | 4 | 6–59 M | 100% | 1,350,759 | no | 1,350,759 | FALSE |
| SSD | Measles | 2020 | 1 | 4 | 6–59 M | 49% | 1,350,759 | no | 659,330 | |
| SSD | Measles | 2023 | 1 | 4 | | 92% | 1,396,213 | no | 1,284,516 | |
| SSD | Measles | 2026 | 1 | 4 | | 92% | 1,465,629 | no | 1,348,379 | |

*Continued on next page*

*Appendix 1—table 17 continued*

| Country | Vaccine | Year | Age_ from | Age_ to | Age range verbatim | Coverage (national level) | Target (national level) | Subnational campaign | Number vaccinated | Affected by covid-19 |
|---|---|---|---|---|---|---|---|---|---|---|
| SSD | Measles | 2029 | 1 | 4 | | 92% | 1,513,497 | no | 1,392,417 | |
| TCD | Measles | 2019 | 1 | 9 | 6M-9Y | 14% | 4,729,086 | yes | 653,511 | |
| TCD | Measles | 2019 | 1 | 9 | 6M-9Y | 4% | 4,729,086 | yes | 210,185 | |
| TCD | Measles | 2019 | 1 | 9 | 6M-9Y | 6% | 4,729,086 | yes | 298,738 | |
| TCD | Measles | 2019 | 1 | 4 | 6–59 M | 21% | 2,259,841 | yes | 467,456 | |
| TCD | Measles | 2020 | 1 | 4 | 6–59 M | 15% | 2,306,276 | yes | 340,046 | FALSE |
| TCD | Measles | 2020 | 1 | 4 | 6–59 M | 2% | 2,306,276 | yes | 43,233 | FALSE |
| TCD | Measles | 2020 | 1 | 4 | 6–59 M | 31% | 2,306,276 | yes | 712,746 | |
| TCD | Measles | 2020 | 1 | 4 | 9–59 M | 100% | 2,306,276 | no | 2,306,276 | |
| TCD | MenA | 2020 | 1 | 8 | 1-8Y | 15% | 4,352,395 | yes | 647,065 | |
| TCD | Measles | 2028 | 1 | 4 | | 82% | 2,681,750 | no | 2,199,035 | |

## Section 2. Figures (appendix)

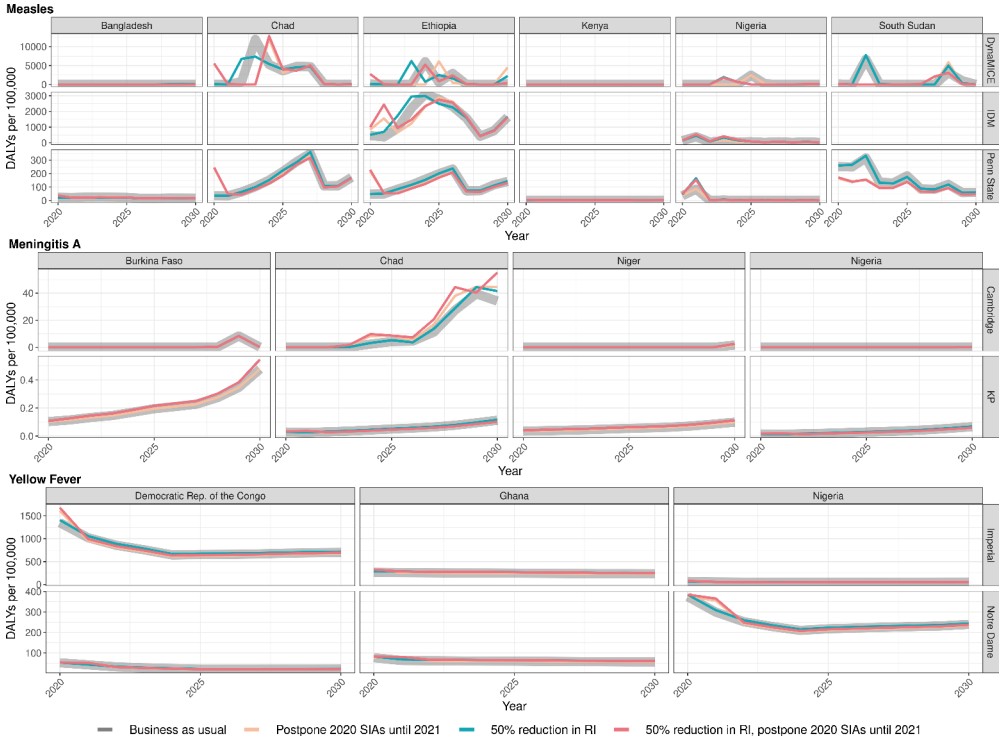

**Appendix 1—figure 1.** Health impact of predicted total disability-adjusted life years for immunisation disruption scenarios and no disruption scenario for measles, meningococcal A, and yellow fever. Model-predicted total disability-adjusted life years (DALYs) per 100,000 population per year for routine immunisation (RI) and campaign immunisation (SIAs – supplementary immunisation activities) disruption scenarios and no disruption scenario (BAU – business-as-usual scenario) for measles, meningococcal A, and yellow fever during 2020–2030.

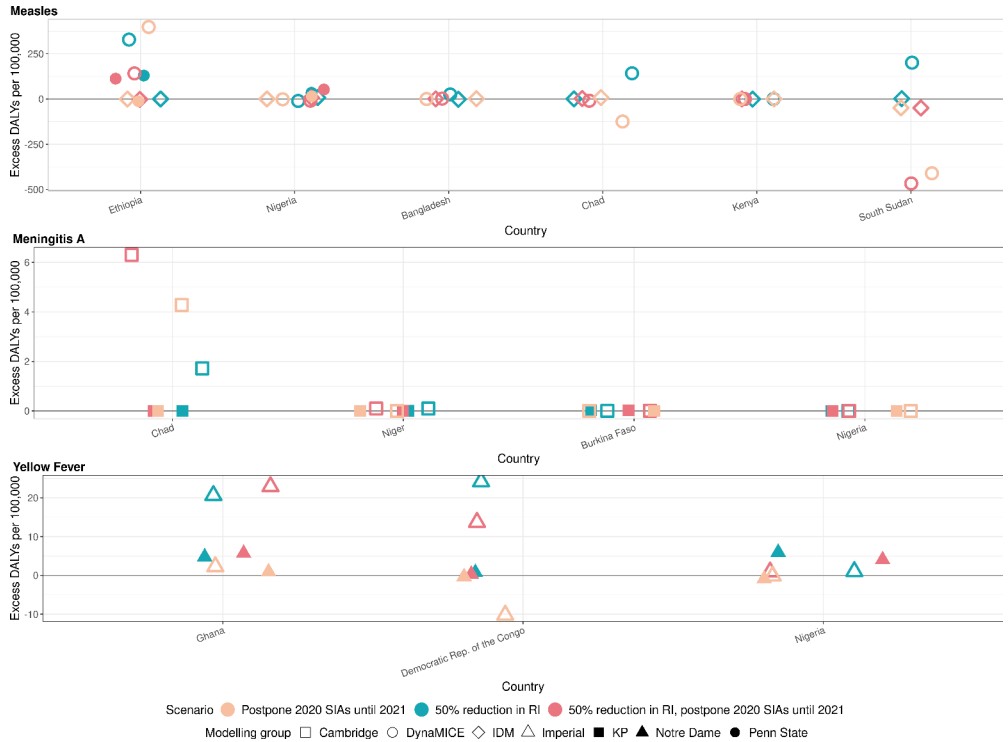

**Appendix 1—figure 2.** Health impact of excess disability-adjusted life years for immunisation disruption scenarios in comparison to no disruption scenario for measles, meningococcal A, and yellow fever. Model-predicted excess disability-adjusted life years (DALYs) per 100,000 population per year for routine immunisation (RI) and campaign immunisation (SIAs – supplementary immunisation activities) disruption scenarios in comparison to no disruption scenario (BAU – business-as-usual scenario) for measles, meningococcal A, and yellow fever. Excess DALYs are summed over 2020–2030.

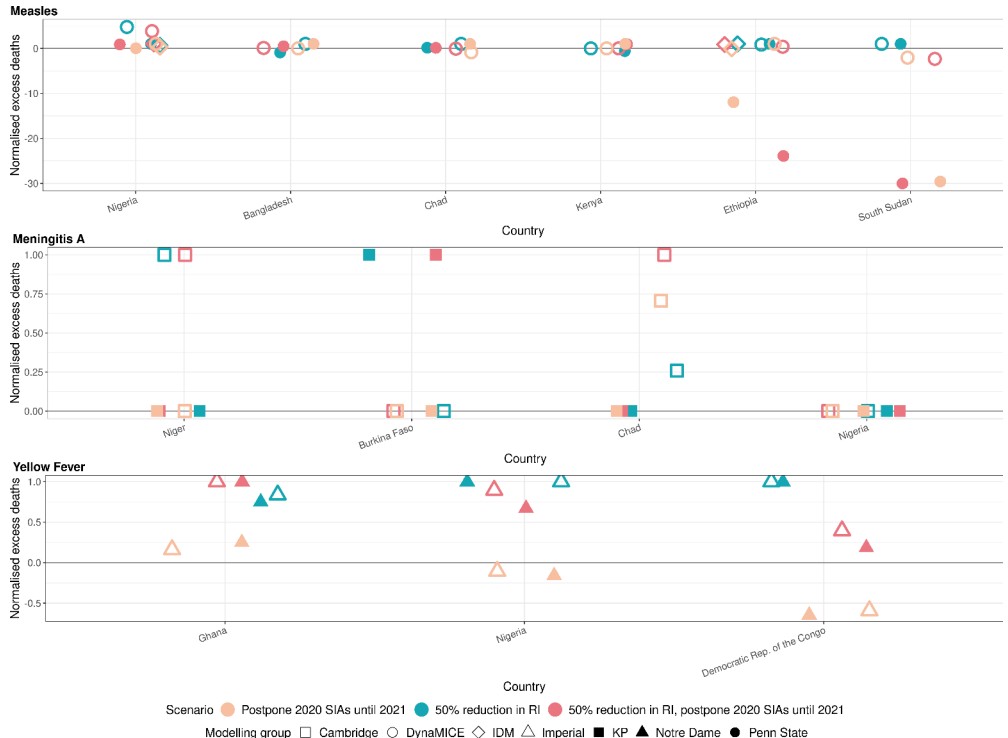

**Appendix 1—figure 3.** Health impact of normalised excess deaths for immunisation disruption scenarios in comparison to no disruption scenario for measles, meningococcal A, and yellow fever. The normalised model-predicted excess deaths per year for routine immunisation (RI) and campaign immunisation (SIAs – supplementary immunisation activities) disruption scenarios in comparison to no disruption scenario (BAU – business-as-usual scenario) for measles, meningococcal A, and yellow fever. Excess deaths are summed over 2020–2030, and the excess deaths are normalised by setting the BAU to 0 and maximum to 1.

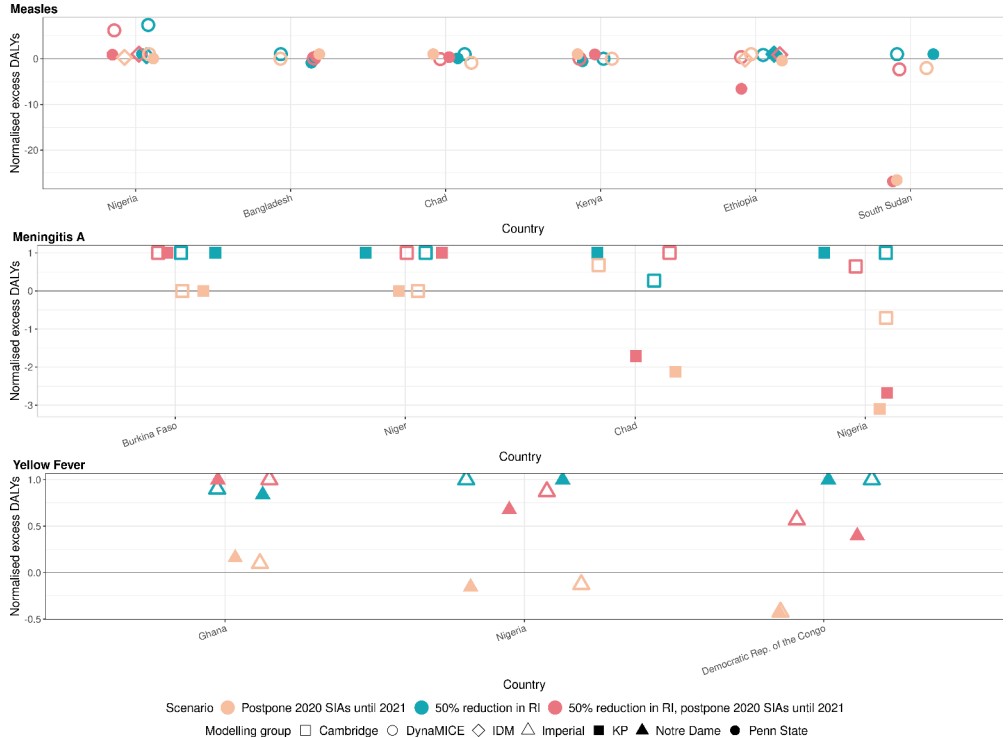

**Appendix 1—figure 4.** Health impact of normalised excess disability-adjusted life years for immunisation disruption scenarios in comparison to no disruption scenario for measles, meningococcal A, and yellow fever. The normalised model-predicted excess disability-adjusted life years (DALYs) per year for routine immunisation (RI) and campaign immunisation (SIAs – supplementary immunisation activities) disruption scenarios in comparison to no disruption scenario (BAU – business-as-usual scenario) for measles, meningococcal A, and yellow fever. Excess DALYs are summed over 2020–2030, and the excess DALYs are normalised by setting the BAU to 0 and maximum to 1.

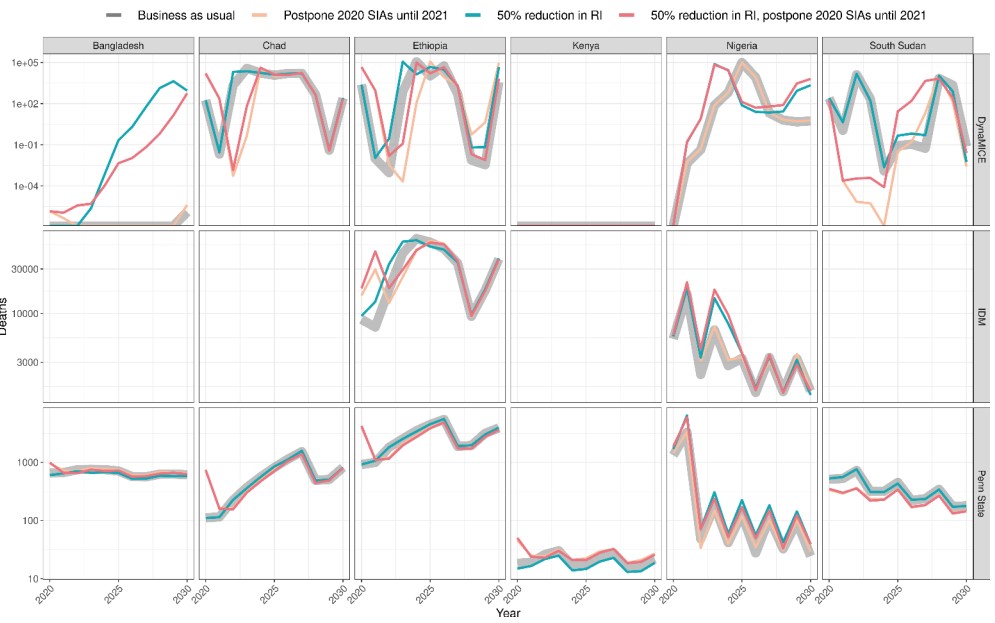

*Appendix 1—figure 5 continued on next page*

*Appendix 1—figure 5 continued*

**Appendix 1—figure 5.** Health impact of predicted total deaths for immunisation disruption scenarios and no disruption scenario for measles. Model-predicted total deaths per year for routine immunisation (RI) and campaign immunisation (SIAs – supplementary immunisation activities) disruption scenarios and no disruption scenario (BAU – business-as-usual scenario) for measles during 2020–2030 per modelling group.

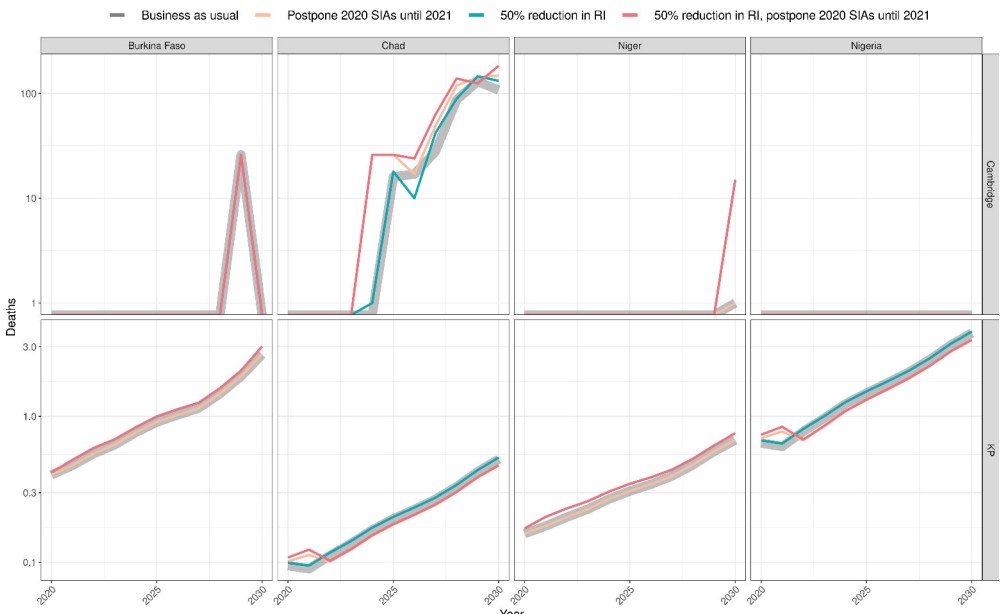

**Appendix 1—figure 6.** Health impact of predicted total deaths for immunisation disruption scenarios and no disruption scenario for meningococcal A. Model-predicted total deaths per year for routine immunisation (RI) and campaign immunisation (SIAs – supplementary immunisation activities) disruption scenarios and no disruption scenario (BAU – business-as-usual scenario) for meningococcal A during 2020–2030 per modelling group.

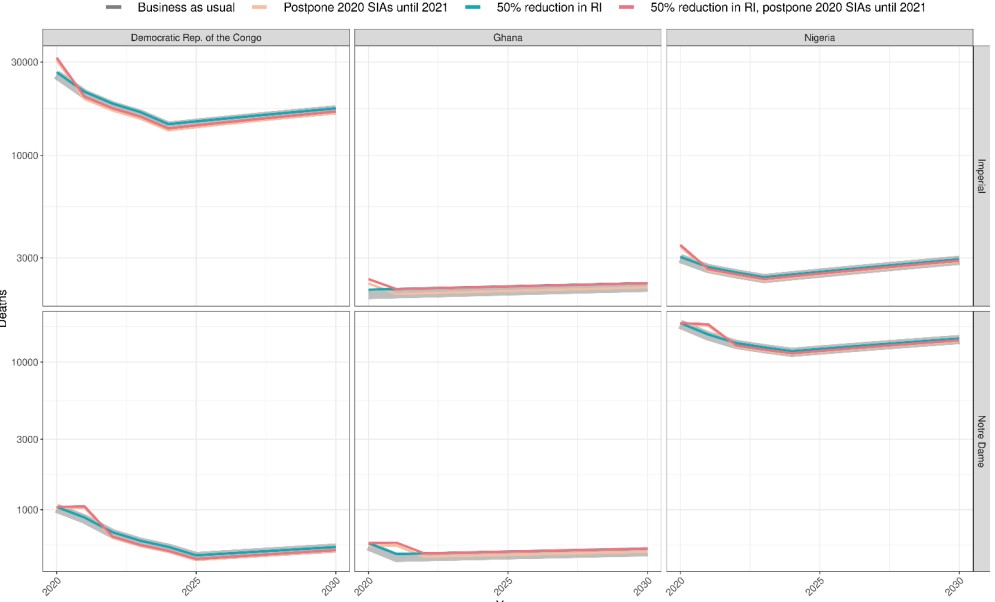

**Appendix 1—figure 7.** Health impact of predicted total deaths for immunisation disruption scenarios and no disruption scenario for yellow fever. Model-predicted total deaths per year for routine immunisation (RI) and campaign immunisation (SIAs – supplementary immunisation activities) disruption scenarios and no disruption scenario (BAU – business-as-usual scenario) for yellow fever during 2020–2030 per modelling group.

## Section 3. Model descriptions

### Measles model - Penn State

The Penn State model is a measles transmission and vaccination model developed at Pennsylvania State University (PSU). It is an age-structured compartmental transmission dynamic model with compartments for susceptible, infected, recovered (due to infection or vaccination) subpopulations. A proportion of infected people will die depending on their age and country characteristics (*Wolfson et al., 2009*), as per the DynaMICE model. The model projects the total number of infections and deaths in 1 year age cohorts, up to age 100 years, in each year according to an annual attack rate that is modeled as a logistic function of the annualised proportion of the population that is susceptible. The slope and intercept of this logistic function, which governs the proportion of available susceptibles that are infected in each year, is fitted independently for each country to observed annual case reporting and vaccination coverage (routine and supplemental campaigns) for each country between 1980 and 2017; for details on the fitting methods see *Eilertson et al., 2019*. Vaccine efficacy for routine immunization is assumed to depend on the age at first dose (9 m or 12 m) as described in *Simons et al., 2012*. The second routine dose is assumed to be preferentially delivered to those children who received the first dose and SIA doses are assumed to be independent of receipt of the first routine dose.

### Measles model - DynaMICE

DynaMICE (DYNAmic Measles Immunisation Calculation Engine) is a measles transmission and vaccination model developed by LSHTM with input from Harvard University and the University of Montreal. It is an age-structured compartmental transmission dynamic model with compartments for maternal immune, susceptible, infected, recovered, and vaccinated subpopulations. A proportion of infected people will die depending on their age and country characteristics (*Wolfson et al., 2009*). The population is also stratified by age with weekly age classes up to age 3 years, and annual age classes thereafter up to 100 years. The force of infection is calculated by combining an age-

dependent social contact matrix from the POLYMOD study (*Mossong et al., 2008*), demographic distribution for each country, and an estimated probability of transmission per contact. The probability of transmission per contact is then estimated from the basic reproduction number of measles using the principal eigenvalue method. Vaccination is incorporated as a pulse function and can be delivered to any age or range of ages and in either routine or campaign delivery. Vaccine efficacy is dependent on age and the number of doses received (*Hughes et al., 2020*). The model has been previously described in detail (*Li et al., 2021*; *Verguet et al., 2015*), and has been validated through comparisons to the Penn State and/or IDM models in at least two previous model comparison exercises.

## Measles models - IDM

The IDM model for Nigeria was built using EMOD – an agent-based stochastic disease transmission model (*Institute for Disease Modeling et al., 2018*). The EMOD software is open-source, and the model and documentation of the EMOD software are available at the IDM website (*IDM, 2020*). The model presented here is a discrete-time (daily time steps), an individual-based form of an MSEIR (maternally protected-susceptible-exposed-infectious-recovered) model. A specific prior application of the EMOD model to measles in Nigeria is described in *Zimmermann et al., 2019*; the model employed here is similar but is structured at a finer spatial scale. The transmission dynamics include seasonality, age-stratified heterogeneous transmission, and spatial metapopulations coupled by migration, the parameters of which have been calibrated to reproduce the seasonality, age-distribution, and spatial correlation of measles cases in Nigeria. Routine vaccination with a first dose is delivered to covered individuals at 9 months of age; the second dose at 12 months; and SIA vaccination is distributed to covered individuals in the target age range in a pulse over the course of 2 weeks; no correlation between the two routine doses or between the routine and SIA doses is assumed.

The IDM model for Ethiopia is a semi-monthly, stochastic, compartmental, measles transmission model. The key model assumption is that measles transmission is determined by a susceptible population and an infectious population, while all other (recovered, deceased, immunised, etc.) populations can be ignored. At a high level, children missed by routine immunization (estimated via coverage and birth rates under vaccine efficacy assumptions similar to those in the Nigeria model) enter the population susceptible to measles, where they can either be infected or be immunised in a vaccination campaign. Transmission is assumed to have annual seasonality with rates estimated for every semi-month of the year via a regression against observed cases accounting for under-reporting as an unknown constant over the model-time period. Both the volatility in the transmission process and the effects of past vaccination campaigns on overall susceptibility are also estimated from the surveillance data. For a detailed example of the model applied to immunizations questions in Pakistan, see *Thakkar et al., 2019*.

## Men A model - Cambridge

The University of Cambridge MenA model is a compartmental transmission dynamic model of *Neisseria meningitidi*s group A (NmA) carriage and disease to investigate the impact of immunisation with a group A meningococcal conjugate vaccine, known as MenAfriVac, as published by *Karachaliou et al., 2015*. The model is age-structured (1 year age groups up to age 100) with continuous ageing between groups. Model parameters were based on the available literature and African data wherever possible, with the model calibrated on an ad-hoc basis as described below.

The population is divided into four states, which represent their status with respect to the meningitis infection. Individuals may be susceptible, carriers, ill or recovered, and in each of these states be vaccinated or unvaccinated, with vaccinated individuals having lower risks of infection (carriage acquisition) and disease (rate of invasion). We assume that both carriers and ill individuals are infectious and can transmit the bacteria to susceptible individuals. The model captures the key features of meningococcal epidemiology, including seasonality, which is implemented by forcing the transmission rate, the extent of which varies stochastically every year.

Since only a small proportion of infected individuals develop the invasive disease, disease-induced deaths are not included in the model. From each compartment, there is a natural death rate from all causes. Carriage prevalence and disease incidence vary with age, and the model

parameterised these distributions using a dataset from Niger *Campagne et al., 1999*; the case:carrier ratio consequently varies with age. The duration of 'natural immunity' is an important driver of disease dynamics in the absence of vaccination but good data on this parameter is lacking; instead, prior estimates are used (*Irving et al., 2012*).

The model assumes that mass vaccination campaigns occur as discrete events whereas routine immunisation takes place continuously. We allowed the duration of protection to vary uniformly between 5 years and 20 years for the 0–4 year-olds and 10–20 years for over 5-year-olds. For the 200 runs, we selected pairs of values for these two parameters so that duration of protection for the older age group is not shorter than the duration of protection for 0–4 year-olds (*White et al., 2019*; *Yaro et al., 2019*). Vaccine efficacy against carriage and disease is 90%.

Disease surveillance is not comprehensive across the meningitis belt, so the disease burden is uncertain in several countries. Therefore, the model classifies the countries into three categories, based on the incidence levels using historical data. This classification defines the transmission dynamic parameters. The model generates estimates of case incidence, to which a 10% case-fatality ratio is applied to estimate mortality (*Lingani et al., 2015*). To estimate DALYs it is assumed that 7·2% of survivors have major disabling sequelae with a disability weight of 0.26 (*Edmond et al., 2010*).

Countries were stratified into high and medium risk, and different infection risks applied based on this stratification. As there was insufficient information to define infection risk on a country-by-country basis, the approach/stratification was agreed upon with experts in the WHO meningitis team. For countries only partly within the meningitis belt, only the (subnational) area at risk was included.

To produce estimates on the impact of vaccination, 200 simulation runs were generated by stochastically varying the baseline transmission rate to reflect between-year climactic or other external variability. Although each individual simulation reflects the reality of irregular and periodic epidemics, as visually compared to time series from Chad and Burkina Faso and analysis of inter-epidemic periods, the resulting averaged estimates give a stable expected burden of disease over time. Uncertainty in other model parameters is currently not quantified.

## Men A model - KP

The KP model for serogroup A Neisseria meningitidis (MenA) was developed at Kaiser Permanente Washington in partnership with the US Centers for Disease Control and Prevention and the Burkina Faso Ministry of Health (*Jackson et al., 2018*; *Tartof et al., 2013*). It is a dynamic, age-structured, stochastic compartmental transmission model, with compartments to represent MenA colonization, disease, and immunity. Natural infection with MenA is assumed to lead to resistance to future colonization and disease, and repeated infections further reduce risk, although protection wanes over time. The age-dependent force of infection ('who acquires infection from whom') matrix varies seasonally to account for differential MenA transmission between dry and rainy seasons. Model parameters, including the force of infection, were estimated using approximate Bayesian calculation, with prior distributions informed by the literature. Mass campaigns occur among persons aged 1–29 years (possibly with catch-up campaigns at the initiation of routine immunization), in which immunization is assumed to occur in the first week of the month during which a campaign is scheduled. Routine immunization is assumed to occur during the first week of the month in which a child reaches 9 months of age.

## Yellow fever model - Imperial College London

The Imperial College London yellow fever model is a static transmission model assuming a constant force of infection (FOI) for each country at risk of YF. It is estimated from YF occurrence data as well as serological data where available. The model also uses environmental covariates, information on vaccination activities, and demographic projections to estimate relative risk and thus transmission intensity for YF. The original framework was developed by *Yellow Fever Expert Committee et al., 2014* and was subsequently extended by *Gaythorpe et al., 2019* and *Gaythorpe et al., 2021b*. The full model description is given in *Gaythorpe et al., 2021b*. The model was estimated within a Bayesian hierarchical framework from serological survey data and outbreak occurrence information

up to the year 2019; it has also been assessed against new serological surveys as they became available, shown in *Jean et al., 2016*.

## Yellow fever model - University of Notre Dame

The University of Notre Dame yellow fever (YFV) model is a static transmission model that assumes a constant force of infection (FOI) for each endemic country (*Perkins et al., 2021*). Yellow fever infections in the human population are thus modeled as spillover events from non-human primates, so human-to-human transmission observed in urban outbreaks is not considered. Accordingly, our model is intended to capture long-term changes in YFV burden on account of changes in vaccination coverage rather than to realistically capture interannual variability due to YFV epizootics in non-human primates and occasional outbreaks in humans.

We calibrated our YFV transmission model to multiple sources of epidemiological data collected in sub-Saharan Africa at the first administrative level subnationally. First, we quantified past exposure to YFV by estimating the force of infection in 23 administrative units using data collected in serological surveys. We then related the predicted number of YFV infections at each of the 23 administrative units to the corresponding reported outbreak data collated by *Yellow Fever Expert Committee et al., 2014* to quantify the extent of underreporting. We then obtained estimates of the total number of infections at each administrative unit in sub-Saharan Africa by relating our estimates of underreporting to the total number of reported cases and deaths in each administrative unit. This allowed us to estimate a posterior distribution of a single FOI for each administrative unit in sub-Saharan Africa. Because the FOIs that we estimated are sensitive to the number of reported cases and deaths, we smoothed across our estimates by performing a regression analysis with spatial covariates. We considered multiple regression models and generated an ensemble prediction by weighting the predicted FOI from each regression model based on performance in ten-fold cross-validation at the country level. National-level FOI estimates were obtained by weighting the ensemble spatial prediction of FOI according to WorldPop 2015 population density estimates at the first administrative level and then summing to obtain national FOIs (*WorldPop, 2016*).

To project the number of yellow fever cases and deaths in each country under a given vaccination coverage scenario, we first scaled the national-level FOI by the proportion of the population that is unvaccinated. We then used the scaled FOI estimate to project the annual number of YFV infections and multiplied this quantity by the probabilities of disease and death reported by *Johansson et al., 2014* to obtain estimates of the annual number of YFV cases and deaths. We assume a 0.975 probability of protection from infection among those who are vaccinated based on *Jean et al., 2016*, with this level of protection assumed to be lifelong based on a single dose. In the event of campaigns, we assume that individuals are vaccinated randomly and irrespective of prior vaccination through another campaign or routine vaccination.

## Section 4: Drivers of model similarities and differences

### Measles

All three measles models (Penn State, DynaMICE, and IDM) are MSRIV (maternally protected, susceptible, infected/infectious, recovered, vaccinated) transmission models. While Penn State and DynaMICE models are age-structured compartmental transmission dynamic models, IDM is an agent-based stochastic disease transmission model. The three models differ in terms of the magnitude of the increased burden they project due to coverage disruptions in 2020, with DynaMICE generally being the most pessimistic (greatest increase in burden) and Penn State generally the most optimistic (smallest increase in burden).

These differences stem particularly from the way vaccine coverage is translated into vaccine impact. DynaMICE directly translates national-level coverage into impact using vaccine efficacy assumptions within an age-dependent mass-action model framework, modified by age at vaccination and whether or not SIA or MCV2 doses go to those who have already received MCV1. Hence any susceptibility gaps that develop as a result of declines in coverage or postponement of SIAs are soon translated into increased numbers of cases.

The Penn State model fits a logistic relationship between annual attack rate and the proportion susceptible in the population independently to each country (methods described in *Eilertson et al.,*

*2019*). The slope and intercept of this function govern how quickly measles cases respond to increases in the proportion susceptible; a steep slope indicates that the probability of infection increases quickly with a small increase in the proportion susceptible (i.e. a large outbreak is likely after a small disruption). The shape of this function is fit to the annual measles time series from 1980 to 2019. If the slope of this function is shallow based on the historical pattern, then a large reduction in coverage (large increase in susceptibles) would be necessary to generate a large and immediate outbreak.

The IDM model uses a similar SIR framework as DynaMICE but is an individual-based model that reflects subnational heterogeneities in dose and disease burden distribution.

### Meningococcal A

The meningococcal disease models (Cambridge, KP) are both stochastic, age-structured, compartmental dynamic transmission models based on the SIR framework. The major structural differences between the models are around (a) how they handle immunity post-infection: where the Cambridge model has waning protection from one immune compartment (in which individuals are completely immune), the KP model assumes a gradient of susceptibility following infection with compartments for high and low immunity; and (b) the duration of vaccine-induced immunity: where the Cambridge model assumes a shorter duration of protection than the KP model. The differences in the results arise mainly because of the differing assumptions about the duration of vaccine protection.

### Yellow fever

The YF models (Imperial, Notre Dame) are both static cohort models which provide annual numbers of infections, cases and deaths given existing vaccination coverage immunity. They follow a similar format in terms of how burden is calculated given force of infection estimates. One difference here is when vaccination is assumed to take effect with the Imperial model showing the influence of vaccination from the beginning of the year and Notre Dame, from the end.

The models differ in how they estimate the force of infection for each province. Both models use serological survey data and outbreak information but the Imperial model uses a larger number of serological studies and only focuses on outbreak occurrence whereas the Notre Dame model also takes into account outbreak size but includes fewer serological studies. Both models use environmental covariates to extrapolate to countries with fewer data but the specific covariates incorporated differ between groups. As a result, the Imperial model generally produces higher estimates of the force of infection except in Nigeria where the force is higher for the Notre Dame model.

## Section 5: Coverage assumptions

These assumptions were determined through consultation with disease and immunisation programme experts across partners at the global, regional, and national levels.

To generate 'business as usual' assumptions for routine immunisations in 2020–2030, we considered historical coverage from WUENIC (WHO and UNICEF Estimates of National Immunization Coverage) for the previous five years. We assumed that MCV1 (measles first dose) coverage stayed at the mean level seen in 2015–19, and that MCV2 (measles second dose) stayed at the highest level seen in 2015–19. Where a country had no MCV2 coverage in the period 2015–19, we assumed that future MCV2 coverage would be 50% of the MCV1 mean coverage for 2015–19. We assumed that yellow fever coverage stayed at the mean level seen in 2015–19. Where a country had no yellow fever coverage in 2015–19, we assumed this stayed constant at the mean level of MCV1 coverage seen in 2015–19. We assumed that coverage of meningitis A stayed at the highest level seen in 2015–19. Where no meningitis A coverage was available for at least one full year, we assumed that future meningitis coverage stayed constant at the mean level of MCV1 coverage seen in 2015–19. However, for countries where meningitis A vaccine was targeted at infants over 15 months, we assumed this matched the highest level of MCV2 coverage seen in 2015–19.

In terms of future vaccine introductions, we assumed that countries would introduce MCV2 and YF in 2022 (where they had not done so already). For meningitis A, all countries considered had already introduced routine immunisation.

Our assumptions about the frequency and coverage level of vaccination campaigns or supplementary immunisation activities (SIA) in 2020–2030 also varied by pathogen. For measles we looked at the historic frequency, that is the interval between the last two prospectively planned national SIAs, and assumed the same frequency in future years. We assumed the same coverage level as in the country's last national-level measles SIA. For yellow fever, we included all completed and planned campaigns (both planned and reactive) in 2019 and 2020, and campaigns recommended in WHO's 2016 Eliminate Yellow Fever (EYE) strategy for the period 2021–2030, assuming 85% coverage of the subnational target population for 2020–2030 (and for 2019 if actual coverage was unavailable). For meningitis A, we included all completed and planned campaigns in 2019 and 2020 (at the actual or forecasted coverage level), but assumed no further campaigns took place from 2021 onwards.

