## [Decision Letter]

**Acceptance summary:**

This study will be of interest for those working in global health and public health officials responsible for immunization against measles, meningococcal A, and measles. An important methodological strength of the study is the use of multiple independently developed transmission models from different teams to estimate the potential disease impacts of vaccination program delays and reductions due to COVID-19 pandemic disruptions. This study highlights the importance of supplementary immunization campaigns in determining future outbreaks of these diseases.

**Decision letter after peer review:**

Thank you for submitting your article "Impact of COVID-19-related disruptions to measles, meningococcal A, and yellow fever vaccination in 10 countries" for consideration by *eLife*. Your article has been reviewed by 3 peer reviewers, one of whom is a member of our Board of Reviewing Editors, and the evaluation has been overseen by a Senior Editor. The following individual involved in review of your submission has agreed to reveal their identity: Trish Campbell (Reviewer #3).

Essential Revisions:

1) It currently impossible to assess inter-model variability in characterization of uncertainty with only model averages and ranges. Please include some results and discussion of inter-model prediction variability in the main text.

2) Please include more details on the statistical methods used to derive the average and min/max ranges.

3) Please comment on validation methods for the models.

4) The impact of reduced transmission due to COVID-19 mitigation measures seem to be missing. For example, measles generally appeared to have the largest impact of a delay, but presumably transmission would also be reduced due to COVID-19 mitigation measures. Some data/modeling/discussion on this would provide important context.

5) Figure 1 does not seem to have the "Business as Usual" line (or BAU simulations?). It is hard to assess and compare these projections without that.

6) There are general labeling issues with the figures and tables that make the paper much harder to read and assess for validity that need to be fixed, see individual reviewer comments.

*Reviewer #1:*

This study models the predicted impact of the COVID-19 pandemic on vaccination programs against three pathogens in multiple countries and the long-term health consequences of disruption to these programs. The study question is timely and important given the potential long-term impacts that disruptions in vaccinations may have on infectious disease mortality, and highlights the importance of reinstating routine and campaign vaccination programs.

The approach used by the authors is to aggregate results from multiple independently developed prediction models, and to present the average and rage of predictions. Cross-model comparisons can be a useful method for making predictions; when models give similar answers, confidence in the results is generally bolstered, and when model results differ, it can point to uncertainties in the disease epidemiology or disease process that help to better understand the range of potential outcomes. In this study, different models made predictions that varied by several orders of magnitude. However, these differences are barely commented on nor explored by the authors. The average of model predictions may not be the most appropriate statistic to aggregate model predictions in this case, because the average is generally driven by whichever model predicted the highest incidence rate. There are some labeling issues on some figures that make the results hard to understand and interpret, as it is not entirely clear what mortality rates would have been without the pandemic. I therefore feel that more work needs to be done to present and contextualize model predictions in order to have confidence in the results.

• When model predictions are very different from each other, I don't think it is appropriate to present the average (ranges) of model predictions as the main result. Figure 1 should probably instead show individual model average predictions rather than the cross-model average prediction to better highlight differences across models. As it is, the average is generally highly influenced by whichever model made the highest predictions and do not give a good measure of the central tendency.

• Please discuss the differences in model predictions in the results and Discussion sections, and provide some explanation as to why this might have occurred. These results are not mentioned at all in the main text and are buried in the appendix.

• If the authors decide to keep the average, we need more information on the statistical methods used to derive this average. Were the results from each model weighted equally? How are different predictions from the same model (different runs/parameter sets) dealt with? What do the minimum and maximum predictions represent in the figures, the minimum and maximum of model averages, or the minimum and maximum across all model simulations?

• Please briefly comment on whether there were broad similarities or important differences across models in the methods section.

• It would be useful to briefly define the parameters of the Business As Usual scenario in the text in terms of vaccination coverage and assumptions.

• Figure 1: this figure is rather baffling and difficult to interpret for various reasons. Firstly, the labels (A,B,C) are not defined in the legend. Secondly, none of the acronyms are defined in the legend (see comment below). Thirdly, some scenarios appear to be missing; for example, the business as usual scenario is not in any of the panels. The orange scenario also appears to be missing for some countries for mysterious reasons. Because there is no business as usual scenario, it is difficult to know what would have been the mortality rate without program disruption, so it is not possible to see what has been the impact of different disruptions. It also seems strange that for some countries, the maximal disruption (red) scenarios appear to lead to lower mortality than less disruptive scenarios (orange and blue).

• Please consider removing most acronyms from the text, figures, and tables. Most acronyms do not substantially shorten the text at the cost of making the text much harder to understand. In most cases it is not necessary as there is plenty of space in figures and tables to fully spell out words. It should not be necessary to have to constantly refer to a table in the appendix to understand what is going on. In general, tables and figures should also be self-sufficient, so if it is absolutely necessary to use acronyms, these should always be defined in the legend (figures) or a footnote (tables).

• There is no reference to Table 1 in the manuscript.

• Table 3: Given the vastly different epidemiology between different countries, I think it would be more useful to present the results by country in this table than to average results across countries; this is because the average across countries does not apply to any individual country; it also does not appear to weight results according to different country population sizes, so is it is not applicable to any region either.

• Many references to tables/figures appear to be incorrect. For example, on P5 a reference is made to Table S11 when I think the table referenced should be Table S16. Please double check all table/figure references. This made the manuscript much harder to follow.

• The information in Table S14 would be more interpretable and useful if it were presented as free text supplementary methods rather than in a table. That way the assumptions and algorithms used to make decisions can be made more detailed and explicit. As it is this table is hard to follow.

• Table S17: it is not clear why vaccination campaigns from 2019 would be affected by the pandemic.

• Supplementary Figures: Please flip supplementary figures so they are in the same direction as the rest of the text to make reading easier. If the authors want to keep the same figure resolution it would be more useful to simply format the page in landscape rather than portrait format.

*Reviewer #2:*

The work is predicated on using multiple models for each pathogen. It states that the models have been validated, but there is no additional information on this. While thorough evidence on validation is almost certainly in the cited papers it would be very helpful to understand that in this context of this manuscript. For example, were all models validated on out-of-sample data from multiple locations and times? Table 1 indicates that some were fitted to data and some were calibrated. Those are fine approaches, but what was done to validate beyond fitting or calibrating to data? Additional comparison between models would be helpful to advance the science of employing multiple models for important use cases like this.

The model projections include multiple sources of uncertainty, yet these are only shown as generalized ranges in the two main figures. Those ranges make it impossible to assess inter-model variability in characterization of uncertainty and changes potential correlation structure that may arise due to seasonality, for example. Moreover, these outcomes are important for a range of decision makers and uncertainty should therefore be characterized throughout, including in the tables, text, and most critically the abstract and discussion.

The impact of reduced transmission due to COVID-19 mitigation measures seem to be missing. For example, measles generally appeared to have the largest impact of a delay, but presumably transmission would also be reduced due to COVID-19 mitigation measures. Some data/modeling/discussion on this would provide important context.

The finding that some delays or reductions were associated with decreased future risk seems to be a function of model structure, not reality. This is potentially confusing and misleading to public health officials and could likely be addressed with updated model structures, parameterization, or synthesis of results.

Figure 1 does not seem to have the "Business as Usual" line (or BAU simulations?). It is hard to assess and compare these projections without that.

Some of the results are not very clear. For example, there doesn't seem to be clear evidence of a predicted measles outbreak in 2025 in Nigeria. How much confidence to the models have in this? The uncertainty range certainly seems large and neither BAU nor 50% RI are shown.

The use of "X deaths per 100,000" is confusing in the context of 2020-2030. It would be helpful to say "yearly" (if I am understanding it correctly).

Line 95: should be pathogens, not antigens

Use country names instead of abbreviations in figures

Figure 1: It would be helpful to have all y-axes being at zero

Figure 1: the panels do not indicate which pathogen they represent

Line 169: Why does the outcome metric change here? Deaths per 100,000 to percent change?

*Reviewer #3:*

In this study, Gaythorpe et al. use a model-based approach to quantify the impact of disruptions to routine vaccination and catch-up campaigns caused by the COVID-19 pandemic. Interruptions to measles, meningococcal A and yellow fever vaccination programs were investigated for several low- and lower-middle income countries where catch-up campaigns had been planned for 2020. Previously-validated disease models were simulated and results averaged for selected diseases in selected countries, with models assuming that routine vaccination coverage and catch-up campaigns would return to normal levels in 2021. The authors found that decreases to routine vaccination coverage and delaying of catch-up campaigns have the potential to cause outbreaks of measles and yellow fever in some countries, while short-term disruption of meningococcal A vaccination is less important.

Strengths

Dynamic transmission models allow the authors to compare the interventions that did occur to the counterfactual scenario of no disruption. The authors evaluate the impacts on disease incidence over a ten-year time frame, sufficient to capture short- to medium-term effects due to the disruption of vaccination programs due to COVID-19. The models used for the study have been previously validated and published, and rather than relying on a single model for each disease, the authors have combined the results from 2-3 models for each disease. The authors' combination of results from multiple models is a useful technique to mitigate structural uncertainty in models arising from different assumptions about the disease transmission process.

The authors are transparent about the considerable heterogeneity found between the results of different models. While the numerical estimates may not be robust, the trends identified across the study were mostly consistent between models. Thus, despite model differences, the study enables the identification of diseases and countries that are likely to fare the worst due to COVID-19 related disruptions to vaccination, enabling action to be taken to prevent outbreaks where necessary.

Weaknesses

Model results were averaged using arithmetic means, placing equal weight on the results of each model. For most of the disease/country combinations examined, results were the arithmetic mean of the results of only two models. For many of these disease/country combinations, the models used produced very disparate results, sometimes in magnitude but also on several occasions the models disagreed as to whether the disruption was beneficial or detrimental. The large differences between some of the model results suggests that one or both models used may not be capturing an important aspect of the transmission process. The work could be extended by averaging over a larger number of models, although this is likely to be quite time consuming, or weighting results based on an assessment of how well the model fits available data.

For yellow fever, the models used were not designed to capture outbreak dynamics and so are not perhaps ideal for use in this study. This appears to be a major limitation of the yellow fever analysis, although it is clearly acknowledged as such by the authors.

As a general comment, I found some of the tense changes throughout the results to be awkward and distracting as they interrupted the flow of the text. Also, the order of the models changes between the tables – for faster comparison, it would be better to use the same order each time.

Measles doses in the IDM are not correlated. It would be worth outlining how you ensured that the correct percentage of the population received two doses. e.g. Ethiopia has 64% coverage for MCV1 and only 31% MCV2. Since doses are not correlated, won't this result in giving MCV2 to previously unvaccinated children, increasing the % with any vaccination? If doses are independent, only 20% will receive both doses, 55% 1 dose and 25% no doses, which differ quite a bit from the actual coverage. Did you adjust the coverage input into the model to ensure you achieved the correct distribution?

Additional comments

Line 112: Spell out WUENIC

Line 142: I'm not sure why the line 'would not lead to increased risk of outbreaks in 2020' is included. Why focus on 2020? Isn't there an outbreak in 2023 due to reduced routine immunisation?

Line 158: For clarity, define what you mean by 'overall' – are you still talking about Chad, or now talking about all settings?

Line 242: Do you have any evidence from your modelling that 'there is a high risk of localised outbreaks in these two states in 2021'? Is this even though the campaign occurred in October 2020?

Lines 248-251: I agree that the risk of importation increases again once COVID-19 restrictions are lifted, but since these restrictions weren't included in the models, I'm not sure of the relevance of pointing this out. The risk of an outbreak without restrictions in place should be that found by the models, shouldn't it?

Line 252: This would be better expressed as 'model averages' rather than 'average models'

Line 264: '…outweighs the excess SARS-CoV-2 infection risk to these age groups…'

Appendix PSU model: Model description mentions Palestine and Kosovo, which are not part of this study.

Figure 1: I can't see the BAU line in the plots even though it is listed in the legend. This makes following the discussion in lines 151 to 155 very difficult and the authors' description needs to be taken at face value. Is this an error?

Table 1: DynaMICE used the R0 from fitted models. R0 tends to be model specific. How can we be sure that the values used were transferable to this model?

---

## [Author Response]

Essential Revisions:1) It currently impossible to assess inter-model variability in characterization of uncertainty with only model averages and ranges. Please include some results and discussion of inter-model prediction variability in the main text.

We have now rewritten the Results and redesigned both figures in the main text so that they show results from all the models separately, rather than averages and ranges. We have also added a detailed discussion of the variability between the models to the text. Please see Reviewer 1 Comment 7 for details.

2) Please include more details on the statistical methods used to derive the average and min/max ranges.

We have now removed any mention of average/max/min across the models following comments from several reviewers. We now show results for each model separately, and now include a detailed discussion of drivers of differences between models. Please see Reviewer 1 Comment 6 for details.

3) Please comment on validation methods for the models.

The models were all independently developed and hence they use different methods of validation, usually including some or all of the following: (i) verification of model inputs and assumptions through expert review, (ii) verification of outputs against empirical data and (iii) corroboration of model conclusions against other models (Weinstein et al.)

We have now expanded the model descriptions in Table 1 and have split this into three parts (Tables 1a-c), to ensure that the validation methods used by each model are clearly stated. Please see Reviewer 2 Comment 1 for details.

Reference

Weinstein MC, Toy EL, Sandberg EA, Neumann PJ, Evans JS, Kuntz KM, Graham JD, Hammitt JK. Modeling for health care and other policy decisions: uses, roles, and validity. Value Health. 2001 Sep-Oct;4(5):348-61. https://doi.org/10.1046/j.1524-4733.2001.45061.x

4) The impact of reduced transmission due to COVID-19 mitigation measures seem to be missing. For example, measles generally appeared to have the largest impact of a delay, but presumably transmission would also be reduced due to COVID-19 mitigation measures. Some data/modeling/discussion on this would provide important context.

We agree that this is a critical area that needs further investigation. The difficulty is that each of the three diseases (measles, yellow fever and meningococcal A) has epidemiology and transmission features that are substantially different from those of COVID-19, so there is not much from the COVID-19 experience that can be transferred.

For *yellow fever*, the majority of transmission is sylvatic rather than person-to-person, so COVID-19 mitigation measures are unlikely to have a major effect on incidence, unless they decrease contact between humans and forest animals.

For *meningococcal A*, we find that even with a decrease in vaccine coverage there is limited potential for outbreaks, so decreased transmission due to COVID-19 non-pharmaceutical interventions will only reinforce this.

For *measles*, there is the potential for non-pharmaceutical interventions to decrease transmission. However, measles is much more transmissible than COVID-19 with R_0_ of around 10-20 rather than 2-5 (Guerra et al., 2017), and transmission is generally concentrated among very young children rather than adults. Hence it is unclear whether interventions designed for COVID-19 (mask wearing, closure of schools, workplaces and retail, travel restrictions etc.) will be able to prevent measles outbreaks. Further, while COVID-19 mitigation measures may temporarily reduce measles transmissibility and outbreak risk from measles immunity gaps, the risk for measles outbreaks will rise rapidly once COVID-19 related contact restrictions are lifted (Mburu et al., 2021), which happens at different rates in different parts of countries.

We have added a brief summary of the discussion above to the Discussion section of the manuscript.

References

Guerra FM, Bolotin S, Lim G, Heffernan J, Deeks SL, Li Y, Crowcroft NS. 2017. The basic reproduction number (R0) of measles: a systematic review. *Lancet Infect Dis***17**:e420–e428. doi:10.1016/S1473-3099(17)30307-9

Mburu CN, Ojal J, Chebet R, Akech D, Karia B, Tuju J, Sigilai A, Abbas K, Jit M, Funk S, Smits G, van Gageldonk PGM, van der Klis FRM, Tabu C, Nokes DJ, LSHTM CMMID COVID-19 Working Group, Scott J, Flasche S, Adetifa I. 2021. The importance of supplementary immunisation activities to prevent measles outbreaks during the COVID-19 pandemic in Kenya. *BMC Med* 19:35. doi:10.1186/s12916-021-01906-9

5) Figure 1 does not seem to have the "Business as Usual" line (or BAU simulations?). It is hard to assess and compare these projections without that.

Thank you, this has now been addressed in the updated figures

6) There are general labeling issues with the figures and tables that make the paper much harder to read and assess for validity that need to be fixed, see individual reviewer comments.

Thank you, this has now been addressed with all figures updated.

Reviewer #1:This study models the predicted impact of the COVID-19 pandemic on vaccination programs against three pathogens in multiple countries and the long-term health consequences of disruption to these programs. The study question is timely and important given the potential long-term impacts that disruptions in vaccinations may have on infectious disease mortality, and highlights the importance of reinstating routine and campaign vaccination programs.The approach used by the authors is to aggregate results from multiple independently developed prediction models, and to present the average and rage of predictions. Cross-model comparisons can be a useful method for making predictions; when models give similar answers, confidence in the results is generally bolstered, and when model results differ, it can point to uncertainties in the disease epidemiology or disease process that help to better understand the range of potential outcomes. In this study, different models made predictions that varied by several orders of magnitude. However, these differences are barely commented on nor explored by the authors. The average of model predictions may not be the most appropriate statistic to aggregate model predictions in this case, because the average is generally driven by whichever model predicted the highest incidence rate.

We have now removed any mention of average/max/min across the models following comments from several reviewers. We now show results for each model separately in both the text and the figures, and include a discussion of drivers of differences between models (see our response below for details).

There are some labeling issues on some figures that make the results hard to understand and interpret, as it is not entirely clear what mortality rates would have been without the pandemic. I therefore feel that more work needs to be done to present and contextualize model predictions in order to have confidence in the results.

Thank you. We have edited all figures to account for labelling issues and better reflect the individual model estimates. This has been accompanied with further explanation in the text as to the model differences for each disease area. We have also made the Business As Usual (BAU) scenario burden more apparent in the figures so it is possible to see the mortality in the absence of pandemic disruption.

• When model predictions are very different from each other, I don't think it is appropriate to present the average (ranges) of model predictions as the main result. Figure 1 should probably instead show individual model average predictions rather than the cross-model average prediction to better highlight differences across models. As it is, the average is generally highly influenced by whichever model made the highest predictions and do not give a good measure of the central tendency.

See response to comment 2. We have indeed removed the average and range of model predictions, and now only report each model separately in both the text and the figures.

• Please discuss the differences in model predictions in the results and Discussion sections, and provide some explanation as to why this might have occurred. These results are not mentioned at all in the main text and are buried in the appendix.

See response to comment 2. We now include a discussion about the differences between model projections.

• If the authors decide to keep the average, we need more information on the statistical methods used to derive this average. Were the results from each model weighted equally? How are different predictions from the same model (different runs/parameter sets) dealt with? What do the minimum and maximum predictions represent in the figures, the minimum and maximum of model averages, or the minimum and maximum across all model simulations?

We have now removed any mention of average/max/min across the models following comments from several reviewers.

• Please briefly comment on whether there were broad similarities or important differences across models in the methods section.

There are important differences between the models that contribute to large variations in projections (sometimes of orders of magnitude). We believe that these variations are important, as they capture genuine structural and data uncertainties around these diseases at a time of change.

Measles. All three measles models (Penn State, DynaMICE, and IDM) are MSRIV (maternally protected, susceptible, infected/infectious, recovered, vaccinated) transmission models. While Penn State and DynaMICE models are age-structured compartmental transmission dynamic models, IDM used an unstructured compartmental model in Ethiopia and an agent-based stochastic disease transmission model in Nigeria. The three models differ in terms of the magnitude of the increased burden they project due to coverage disruptions in 2020, with DynaMICE generally being the most pessimistic (greatest increase in burden) and Penn State generally the most optimistic (smallest increase in burden).

These differences stem particularly from the way vaccine coverage is translated into vaccine impact. DynaMICE directly translates national-level coverage into impact using vaccine efficacy assumptions within an age-dependent mass-action model framework, modified by age at vaccination and whether or not SIA or MCV2 doses go to those who have already received MCV1. Hence any susceptibility gaps that develop as a result of declines in coverage or postponement of SIAs are soon translated into increased numbers of cases.

Both IDM models use a similar mechanistic framework as DynaMICE, but, particularly in the application to Nigeria, IDM used an individual-based model that reflects subnational heterogeneities in dose and disease burden distribution.

The Penn State model fits a logistic relationship between annual attack rate and the proportion susceptible in the population independently to each country (methods described in Eilertson et al. 2019 https://doi.org/10.1002/sim.8290). The slope and intercept of this function govern how quickly measles cases respond to increases in the proportion susceptible; a steep slope indicates that the probability of infection increases quickly with a small increase in the proportion susceptible (i.e. a large outbreak is likely after a small disruption). The shape of this function is fit to the annual measles time series from 1980-2019. If the slope of this function is shallow based on the historical pattern, then a large reduction in coverage (large increase in susceptibles) would be necessary to generate a large and immediate outbreak.

Meningococcal A. The meningococcal disease models (Cambridge, KP) are both stochastic, age-structured, compartmental dynamic transmission models based on the SIR framework. The major structural differences between the models are around (a) how they handle immunity post-infection: where the Cambridge model has waning protection from one immune compartment (in which individuals are completely immune), the KP model assumes a gradient of susceptibility following infection with compartments for high and low immunity; and (b) the duration of vaccine-induced immunity: where the Cambridge model assumes a shorter duration of protection than the KP model. The differences in the results arise mainly because of the differing assumptions about the duration of vaccine protection.

Yellow Fever. The YF models (Imperial, Notre Dame) are both static cohort models which provide annual numbers of infections, cases and deaths given existing vaccination coverage immunity. They follow a similar format in terms of how burden is calculated given force of infection estimates. One difference here is when vaccination is assumed to take effect with the Imperial model showing the influence of vaccination from the beginning of the year and Notre Dame, from the end.

The models differ in how they estimate the force of infection for each province. Both models use serological survey data and outbreak information but the Imperial model uses a larger number of serological studies and only focuses on outbreak occurrence whereas the Notre Dame model also takes into account outbreak size but includes fewer serological studies. Both models use environmental covariates to extrapolate to countries with fewer data but the specific covariates incorporated differ between groups. As a result, the Imperial model generally produces higher estimates of the force of infection except in Nigeria where the force is higher for the Notre Dame model.

As the explanation of the drivers of differences between model predictions is quite detailed and lengthy, we have added it to Appendix Section 4, with a reference to this in the Methods.

• It would be useful to briefly define the parameters of the Business As Usual scenario in the text in terms of vaccination coverage and assumptions.

Thank you for the comment. We do define the assumptions used for this scenario in Appendix Section 1 Table S14, and the actual parameters in Appendix Section 1 Table S16. We would be happy to provide more details. In the meantime, we have added the following text to the Methods to summarise how we developed the assumptions:

“Assumptions for our counterfactual “business as usual” scenario were determined through consultation with disease and immunisation programme experts across partners at the global, regional, and national levels. All assumptions varied by antigen. For routine immunisation, assumptions about future coverage levels were based on historical coverage from WUENIC for 2015-19. For vaccination campaigns or supplementary immunisation activities (SIA), assumptions about future campaigns were based either on patterns of past campaigns or campaigns recommended by WHO.”

More details are given in response below.

• Figure 1: this figure is rather baffling and difficult to interpret for various reasons. Firstly, the labels (A,B,C) are not defined in the legend. Secondly, none of the acronyms are defined in the legend (see comment below). Thirdly, some scenarios appear to be missing; for example, the business as usual scenario is not in any of the panels. The orange scenario also appears to be missing for some countries for mysterious reasons. Because there is no business as usual scenario, it is difficult to know what would have been the mortality rate without program disruption, so it is not possible to see what has been the impact of different disruptions. It also seems strange that for some countries, the maximal disruption (red) scenarios appear to lead to lower mortality than less disruptive scenarios (orange and blue).

Thank you for your feedback on Figure 1. We have relabelled the facets to read Measles, Yellow fever and Meningitis A rather than A,B,C. We have also expanded the acronyms and checked that all scenarios are present and visible. Particularly, we have made the Business As Usual scenario bolder so that the line is more clearly visible. Finally, we have included the individual model results separately so it is possible to see the differences and similarities.

• Please consider removing most acronyms from the text, figures, and tables. Most acronyms do not substantially shorten the text at the cost of making the text much harder to understand. In most cases it is not necessary as there is plenty of space in figures and tables to fully spell out words. It should not be necessary to have to constantly refer to a table in the appendix to understand what is going on. In general, tables and figures should also be self-sufficient, so if it is absolutely necessary to use acronyms, these should always be defined in the legend (figures) or a footnote (tables).

We have now removed all abbreviations/acronyms from the text except for those that are widely used (e.g. WHO, DALY, COVID-19).

• There is no reference to Table 1 in the manuscript.

Thank you, we have added a reference to this table at the beginning of the methods section. Table 1 is now split into three parts (Tables 1a-c)

• Table 3: Given the vastly different epidemiology between different countries, I think it would be more useful to present the results by country in this table than to average results across countries; this is because the average across countries does not apply to any individual country; it also does not appear to weight results according to different country population sizes, so is it is not applicable to any region either.

Thank you, this information is available in Tables S2, S4 and S6 for Measles, Men A and YF, respectively.

• Many references to tables/figures appear to be incorrect. For example, on P5 a reference is made to Table S11 when I think the table referenced should be Table S16. Please double check all table/figure references. This made the manuscript much harder to follow.

Thank you; we have corrected this and checked all table/figure references.

• The information in Table S14 would be more interpretable and useful if it were presented as free text supplementary methods rather than in a table. That way the assumptions and algorithms used to make decisions can be made more detailed and explicit. As it is this table is hard to follow.

We think it might still be useful to keep Table S14 to make it easier to compare across diseases, but we have also added more detailed descriptions as free text in Appendix 5.

“These assumptions were determined through consultation with disease and immunisation programme experts across partners at the global, regional, and national levels.

To generate “business as usual” assumptions for routine immunisations in 2020-2030, we considered historical coverage from WUENIC for the previous five years. We assumed that MCV1 (measles first dose) coverage stayed at the mean level seen in 2015-19, and that MCV2 (measles second dose) stayed at the highest level seen in 2015-19. Where a country had no MCV2 coverage in the period 2015-19, we assumed that future MCV2 coverage would be 50% of the MCV1 mean coverage for 2015-19. We assumed that yellow fever coverage stayed at the mean level seen in 2015-19. Where a country had no yellow fever coverage in 2015-19, we assumed this stayed constant at the mean level of MCV1 coverage seen in 2015-19. We assumed that coverage of meningitis A stayed at the highest level seen in 2015-19. Where no meningitis A coverage was available for at least one full year, we assumed that future meningitis coverage stayed constant at the mean level of MCV1 coverage seen in 2015-19. However, for countries where meningitis A vaccine was targeted at infants over 15 months, we assumed this matched the highest level of MCV2 coverage seen in 2015-19.

In terms of future vaccine introductions, we assumed that countries would introduce MCV2 and YF in 2022 (where they had not done so already). For meningitis A, all countries considered had already introduced routine immunisation.

Our assumptions about the frequency and coverage level of vaccination campaigns or supplementary immunisation activities (SIA) in 2020-2030 also varied by antigen. For measles we looked at the historic frequency, i.e. the interval between the last two prospectively planned national SIAs, and assumed the same frequency in future years. We assumed the same coverage level as in the country’s last national-level measles SIA. For yellow fever, we included all completed and planned campaigns (both planned and reactive) in 2019 and 2020, and campaigns recommended in WHO’s 2016 Eliminate Yellow Fever (EYE) strategy for the period 2021-2030, assuming 85% coverage of the subnational target population for 2020-2030 (and for 2019 if actual coverage was unavailable). For meningitis A, we included all completed and planned campaigns in 2019 and 2020 (at the actual or forecasted coverage level), but assumed no further campaigns took place from 2021 onwards.”

• Table S17: it is not clear why vaccination campaigns from 2019 would be affected by the pandemic.

We indeed do not expect 2019 coverage to be affected by the pandemic – we simply make a simplified assumption of 85% coverage if data are not yet available on 2019 campaigns.

• Supplementary Figures: Please flip supplementary figures so they are in the same direction as the rest of the text to make reading easier. If the authors want to keep the same figure resolution it would be more useful to simply format the page in landscape rather than portrait format.

Thank you, we have rotated the figures and will adhere to the journal formatting requirements for the final submission.

Reviewer #2:The work is predicated on using multiple models for each pathogen. It states that the models have been validated, but there is no additional information on this. While thorough evidence on validation is almost certainly in the cited papers it would be very helpful to understand that in this context of this manuscript. For example, were all models validated on out-of-sample data from multiple locations and times? Table 1 indicates that some were fitted to data and some were calibrated. Those are fine approaches, but what was done to validate beyond fitting or calibrating to data? Additional comparison between models would be helpful to advance the science of employing multiple models for important use cases like this.

The models were all independently developed and hence they use different methods of validation, usually including some or all of the following: (i) verification of model inputs and assumptions through expert review, (ii) verification of outputs against empirical data and (iii) corroboration of model conclusions against other models (Weinstein et al. https://doi.org/10.1046/j.1524-4733.2001.45061.x).

We have now expanded Table 1 (split into Tables 1a-c) and the model descriptions in the Appendix Section 3 to ensure that the validation methods used by each model are clearly stated.

The model projections include multiple sources of uncertainty, yet these are only shown as generalized ranges in the two main figures. Those ranges make it impossible to assess inter-model variability in characterization of uncertainty and changes potential correlation structure that may arise due to seasonality, for example. Moreover, these outcomes are important for a range of decision makers and uncertainty should therefore be characterized throughout, including in the tables, text, and most critically the abstract and discussion.

Thank you; we have now revised both the Results text and the figures so that they show the model projections independently to enable comparisons. We agree that there are many sources of uncertainty that could affect the model results; we now include a detailed discussion of this in Appendix Section 4. Due to the annual nature of some of the models, such as for yellow fever, the influence of seasonality will not be apparent in these results.

The impact of reduced transmission due to COVID-19 mitigation measures seem to be missing. For example, measles generally appeared to have the largest impact of a delay, but presumably transmission would also be reduced due to COVID-19 mitigation measures. Some data/modeling/discussion on this would provide important context.

This is indeed an important issue. Please see Editors’ Comment 4 for our response. While we have not been able to model this directly, we have added discussion to the manuscript on this issue.

The finding that some delays or reductions were associated with decreased future risk seems to be a function of model structure, not reality. This is potentially confusing and misleading to public health officials and could likely be addressed with updated model structures, parameterization, or synthesis of results.

This is indeed an important point to highlight. We think it is not purely an issue of model structure or parameterisation, but also relates to how the results should be interpreted. In particular, decision makers may want to take into account other contextual or programmatic factors besides model results; they may also not wish to delay a campaign simply for the mathematical benefit of being able to vaccinate more children in the future.

We discuss this important issue in the Discussion section:

"In some of our modelled scenarios, postponement of immunisation campaigns does not appear to increase overall cases, if the delay time-period is less than the interval to the next outbreak. Such a scenario is inferred in the immunisation disruption scenarios for postponement of measles campaigns for South Sudan. This does not imply that a postponement is preferred, as we do not take into account other contextual or programmatic factors; rather it reflects the effectiveness of campaigns in closing the immunity gaps and the demographic effect of including more children in delayed campaigns. In instances with very low routine immunisation coverage, there is a possibility that the vaccination campaign is the main opportunity for missed children to be vaccinated. Thus for the same proportion of the same age group targeted by campaigns, more children will be vaccinated for the same coverage levels in countries with birth rates increasing over time. While these results may be useful in the COVID-19 context, there is also considerable uncertainty around both model findings and data inputs such as incidence and vaccine coverage that prohibits further general comment on the optimal timing of campaigns."

Figure 1 does not seem to have the "Business as Usual" line (or BAU simulations?). It is hard to assess and compare these projections without that.

Thank you, we have updated the figure to show this.

Some of the results are not very clear. For example, there doesn't seem to be clear evidence of a predicted measles outbreak in 2025 in Nigeria. How much confidence to the models have in this? The uncertainty range certainly seems large and neither BAU nor 50% RI are shown.

This is an important point about the limitations of future model projections given data and model structure uncertainties, as well as the importance of using multiple models. We do not think that the model results should be interpreted precisely e.g. to indicate that there will definitely be a measles outbreak in 2025 in Nigeria. Indeed, only one of the three measles models predicts an outbreak in 2025, even though all three predict that there will likely be an outbreak (whose size varies depending on the model) in the next few years. This points to the fact that there is a susceptibility gap in Nigeria across all three models, lending greater confidence in that finding.

We have added additional information in Appendix 4 to explain the differences between the models. We now also add the following text to the first paragraph of the Discussion to highlight this:

“However, model projections about future outbreaks differ between models in terms of both timing and magnitude. These differences capture uncertainty around data and model structure that differ between models.”

We have also updated the figures to show the Business As Usual scenario more clearly and to separate the results between models.

The use of "X deaths per 100,000" is confusing in the context of 2020-2030. It would be helpful to say "yearly" (if I am understanding it correctly).

We have changed all references to “X deaths per 100,000 per year” when annual incidence is meant and “X deaths per 100,000 over 2020-2030” when cumulative incidence over the time period is meant.

Line 95: should be pathogens, not antigens

We have updated the sentence:

“To address this, we used transmission dynamic models to project alternative scenarios about postponing vaccination campaigns alongside disruption of routine immunisation, for three pathogens with high outbreak potential and for which mass vaccination campaigns are a key delivery mode alongside routine immunisation – measles, meningococcal A, and yellow fever.”

Use country names instead of abbreviations in figures

This has been updated.

Figure 1: It would be helpful to have all y-axes being at zero

Thank you, we have incorporated this suggestion.

Figure 1: the panels do not indicate which pathogen they represent

Thank you, we have relabelled them.

Line 169: Why does the outcome metric change here? Deaths per 100,000 to percent change?

The outcome metric is generally reported as deaths per 100K; however, for Meningitis A the values for very small, therefore to show the change, the proportion in the form of percentages was used.

Reviewer #3:[…]WeaknessesModel results were averaged using arithmetic means, placing equal weight on the results of each model. For most of the disease/country combinations examined, results were the arithmetic mean of the results of only two models. For many of these disease/country combinations, the models used produced very disparate results, sometimes in magnitude but also on several occasions the models disagreed as to whether the disruption was beneficial or detrimental. The large differences between some of the model results suggests that one or both models used may not be capturing an important aspect of the transmission process. The work could be extended by averaging over a larger number of models, although this is likely to be quite time consuming, or weighting results based on an assessment of how well the model fits available data.

We have now removed any mention of average/max/min across the models following comments from several reviewers. We now show results for each model separately in both the text and the figures, and have included a discussion of drivers of differences between models in Appendix Section 4.

For yellow fever, the models used were not designed to capture outbreak dynamics and so are not perhaps ideal for use in this study. This appears to be a major limitation of the yellow fever analysis, although it is clearly acknowledged as such by the authors.

The models used in this analysis and for the projection of vaccine impact as part of the VIMC focus on long-term estimations of disease burden. The burden of yellow fever is largely driven by sylvatic spillover events; although the urban transmission cycle can lead to explosive outbreaks, the work of the Eliminate Yellow Fever Epidemics (EYE) strategy, countries and Gavi aims to reduce the chance of urban outbreaks. As such, whilst the models will not account for stochastic outbreaks in these terms, they can describe average projected burden on the longer-term. We noted these limitations in the Discussion:

“The models used in this analysis, in particular for yellow fever, are best suited to capture long-term changes in disease burden due to vaccination and cannot capture outbreak dynamics that may arise in the short-term.”

As a general comment, I found some of the tense changes throughout the results to be awkward and distracting as they interrupted the flow of the text. Also, the order of the models changes between the tables – for faster comparison, it would be better to use the same order each time.

We have now standardised the order of the models in each table. We have also checked the tenses used and updated these where necessary.

Measles doses in the IDM are not correlated. It would be worth outlining how you ensured that the correct percentage of the population received two doses. e.g. Ethiopia has 64% coverage for MCV1 and only 31% MCV2. Since doses are not correlated, won't this result in giving MCV2 to previously unvaccinated children, increasing the % with any vaccination? If doses are independent, only 20% will receive both doses, 55% 1 dose and 25% no doses, which differ quite a bit from the actual coverage. Did you adjust the coverage input into the model to ensure you achieved the correct distribution?

Some clarifications about the model structure have been made in Table 1a. Thank you for pointing out this statement, the statement that the two doses are uncorrelated was a misstatement and does not reflect the behavior of the IDM models.

IDM’s models of Nigeria and Ethiopia are structurally different. The Nigeria model is agent-based and spatially resolved at subnational level, and MCV2 is explicitly only given to the recipients of MCV1. IDM’s Ethiopia model, in contrast, is a stochastic compartmental model, but again MCV2 is assumed to go only to recipients of MCV1. The compartmental structure of the Ethiopia model does not separately track 1- and 2-dose individuals explicitly; the impact of MCV1 and MCV2 on the influx of new susceptible persons is described in Equation 1 of the supplement to (Thakkar et al., 2019):Bt=Bt‾(1−0.9V1,t(1−V2,t)−0.99V1,tV2,t)where Bt is the influx of susceptibles, Bt‾ is the total new births, V1,t is MCV1 coverage, and V2,t is MCV2 coverage.

Note that we now explicitly separate out IDM’s models for Nigeria and Ethiopia due to their differences.

Additional commentsLine 112: Spell out WUENIC

We have now spelt out WUENIC:

“Models used routine and campaign vaccination coverage from WUENIC (WHO and UNICEF Estimates of National Immunization Coverage) and post campaign surveys for 2000-2019 …”

Line 142: I'm not sure why the line 'would not lead to increased risk of outbreaks in 2020' is included. Why focus on 2020? Isn't there an outbreak in 2023 due to reduced routine immunisation?

We have now modified the wording as we now focus on results from each model rather than the average result across the models. We also remove the focus on 2020 specifically. The new text reads: “For Ethiopia, a reduction in routine coverage is predicted to lead to outbreaks sooner and increases in overall deaths in all three models (DynaMICE, Penn State models and IDM), while postponing the 2020 campaign only increases deaths in the DynaMICE model.”

Line 158: For clarity, define what you mean by 'overall' – are you still talking about Chad, or now talking about all settings?

We have now modified this to read: “For Chad, both DynaMICE and Penn State models predict an overall increase in deaths with routine coverage drops, but only the Penn State model predicts an increase with a postponement of campaigns.” We have taken out references to outbreaks in 2020 as per Comment 8.

Line 242: Do you have any evidence from your modelling that 'there is a high risk of localised outbreaks in these two states in 2021'? Is this even though the campaign occurred in October 2020?

The models do not indicate a high risk of localised outbreaks after the campaign was successfully run in October 2020. At the time that these model results were originally obtained, the October 2020 campaigns had not yet been executed and were at high risk of being delayed into 2021. Because these two states had not been included in the 2019 campaign as intended, subnational modeling indicated that the long buildup of local susceptibility placed them at high risk of localised outbreaks in the event that campaigns were further delayed past the 2021 high season (Dec 2020-Mar 2021). The text in the manuscript has been changed to clarify this point and is copied below:

“The measles immunisation campaigns for 2020 in Nigeria were specifically targeted at Kogi and Niger states, states that were originally scheduled for inclusion in the campaigns for 2019 across northern Nigeria which were delayed for other reasons. Given the localised build-up of susceptibility in these two states due to low routine immunisation coverage and the long window between campaigns, IDM’s subnational Nigeria model indicated that further campaign delays would result in a high risk of localised outbreaks in these states (one potential explanation for the IDM model predicting worse consequences of delays in these campaigns than the other models). Campaigns targeted specifically to these two states were implemented in October 2020.”

Lines 248-251: I agree that the risk of importation increases again once COVID-19 restrictions are lifted, but since these restrictions weren't included in the models, I'm not sure of the relevance of pointing this out. The risk of an outbreak without restrictions in place should be that found by the models, shouldn't it?

We included it as part of a paragraph on limitations, pointing it out as a further reason to implement postponed campaigns that we did not model. We have revised the section slightly to make it clear that we do not model it: “Further, our models did not include the possibility that COVID-19 restrictions may have temporarily reduced measles transmissibility and the risk of measles outbreaks due to reduced chance of introduction of infection into populations with immunity gaps. This risk rises again rapidly once travel restrictions and physical distancing are relaxed. This is an additional reason (which we do not model) for implementing postponed immunisation campaigns at the earliest opportunity to prevent measles outbreaks as COVID-19 restrictions are lifted (Mburu et al., 2021).”

Line 252: This would be better expressed as 'model averages' rather than 'average models'

We have now removed the use of model averages from the manuscript following comments from several reviewers.

Line 264: '…outweighs the excess SARS-CoV-2 infection risk to these age groups…'

We have updated the sentence:

“Since children and younger age-group individuals are at relatively lower risk of morbidity and mortality from COVID-19 in comparison to elderly populations, the health benefits of sustaining measles, meningococcal A, and yellow fever immunisation programmes during the COVID-19 pandemic outweigh the excess SARS-CoV-2 infection risk to these age groups that are associated with vaccination service delivery points. “

Appendix PSU model: Model description mentions Palestine and Kosovo, which are not part of this study.

Thank you for pointing it out – this was included inadvertently. We have now removed the sentence.

Figure 1: I can't see the BAU line in the plots even though it is listed in the legend. This makes following the discussion in lines 151 to 155 very difficult and the authors' description needs to be taken at face value. Is this an error?

Thank you for highlighting this, we have ensured it is now visible.

Table 1: DynaMICE used the R0 from fitted models. R0 tends to be model specific. How can we be sure that the values used were transferable to this model?

The basic reproduction number R_0_ should in theory be a property of the pathogen and the population rather than the model, since it is an inherent biological/behavioural parameter. While there are many ways of estimating it, in principle these are different ways of measuring the same underlying quality, even though in practice they may differ due to different levels of precision, use of data sources and underlying assumptions. See for instance Heffermann et al. (https://dx.doi.org/10.1098%2Frsif.2005.0042).

So while we agree that R_0_ estimated from one model used in another introduces additional imprecision, we believe that it is theoretically correct to use R_0_ in different situations.

To highlight the limitation, we have added the following sentence to the Discussion: “A key strength of our analysis is that we used 2-3 models for each infection, which allowed investigation of whether projections were sensitive to model structure and assumptions. Each model had different strengths and limitations. For instance, some models measured epidemic properties like reproduction numbers directly, while other models used estimates from other studies.”